# Cartography of Genomic Interactions Enables Deep Analysis of Single-Cell Expression Data

**Md Tauhidul Islam** [1] **& Lei Xing** [1]✉

Remarkable advances in single cell genomics have presented unique challenges and opportunities for interrogating a wealth of biomedical inquiries. High dimensional genomic data are inherently complex because of intertwined relationships among the genes. Existing methods, including emerging deep learning-based approaches, do not consider the underlying biological characteristics during data processing, which greatly compromises the performance of data analysis and hinders the maximal utilization of state-of-the-art genomic techniques. In this work, we develop an entropy-based cartography strategy to contrive the high dimensional gene expression data into a configured image format, referred to as genomap, with explicit integration of the genomic interactions. This unique cartography casts the gene-gene interactions into the spatial configuration of genomaps and enables us to extract the deep genomic interaction features and discover underlying discriminative patterns of the data. We show that, for a wide variety of applications (cell clustering and recognition, gene signature extraction, single cell data integration, cellular trajectory analysis, dimensionality reduction, and visualization), the proposed approach drastically improves the accuracies of data analyses as compared to the state-of-the-art techniques.

Recent advances in high-throughput genomic techniques have led to profound new discoveries in biomedicine[1–6]. Alongside these breakthroughs, how to accurately discern patterns from the high dimensional (HD) and large-scale gene expression data presents a ubiquitous challenge across applications[7–10]. Up to now, analytical techniques, such as dimensionality reduction, discriminant analysis, Bayesian classification, decision-tree, and neural networks, have been used to process the HD data and build predictive models for various tasks such as classification and regression[11–16]. Unfortunately, these techniques fall short in extracting the most discriminative features and often yield sub-optimal results in various biomedical applications.

Biologically, it has long been recognized that gene-gene interactions play a significant role in various cellular processes and provide a unique basis to discriminate cell types and states[17–19]. These interactions have not, however, been utilized explicitly for discovering distinct patterns in genomic systems. An underlying challenge in integrating the information into the data processing pipeline arises from the way that the measured gene expression data are organized. In practice, a vector or matrix (Fig. 1a) represents a convenient but not the most informative way to present a set of gene expression data. When the data are presented in this format, the information of gene-gene interactions are buried in the expression matrix, which disables us from gaining comprehensive insights into genomic interplays and utilizing the information to facilitate downstream data analyses. In this work, we introduce the concept of genomap and provide a cartographic framework to enable the extraction of genomic interaction features for high performance data analysis. We configure the genomap of a cell by placing individual genes into a two-dimensional (2D) grid based on the attributes of their interactions in high dimension. To a certain extent, the construction of a genomap here is analogous to the reconstruction of biomedical images (such as CT and MRI) from

[1]Department of Radiation Oncology, Stanford University, Stanford, California 94305, USA. ✉e-mail: lei@stanford.edu

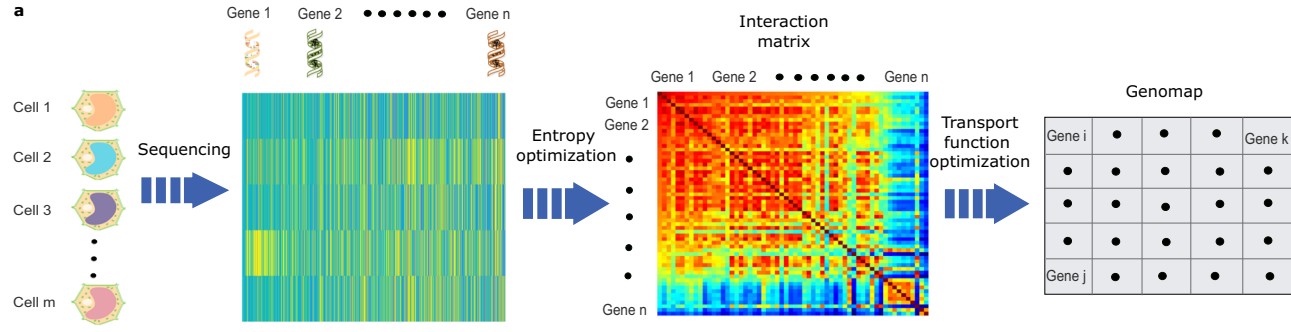

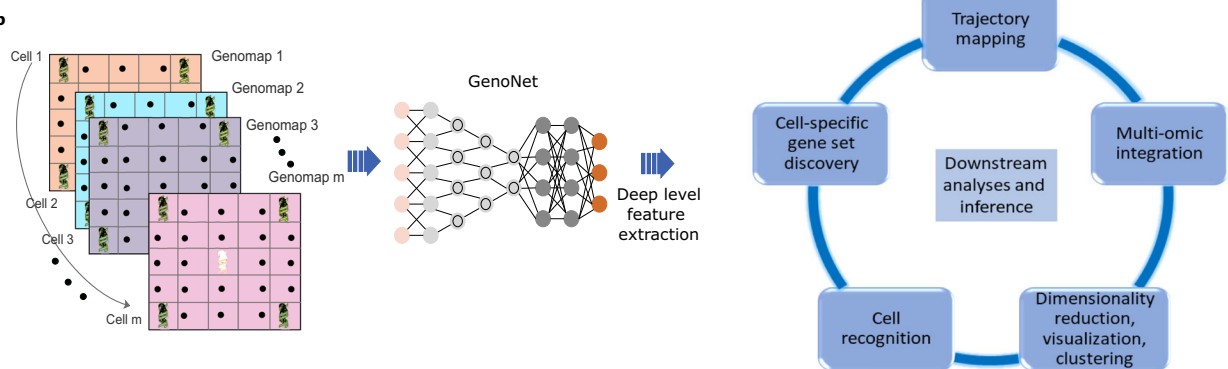

**Fig. 1 | Deep analysis of scRNA-seq data by using genomap and genoNet.**
**a** Workflow of genomap generation from scRNA-seq data. Here, $i \neq j \neq k$, $i = 1, ...,$
$n, j = 1, ..., n,$ , $k = 1, ..., n$. Note that the genomap is dataset dependent and the gene

distribution in the genomaps vary with dataset. **b** GenoNet is applied on the gen-
omaps to extract deep level features for decision-making.

unconfigured sensory data, in which the measured data are trans-
formed into semantically meaningful images that can be better
perceived by a human or computer agent[20,21]. We demonstrate that,
with the introduction of the genomap, the correlations in genomic
data can be exploited effectively by deep learning to greatly facil-
itate the downstream applications.

In the proposed cartography, we first compute the gene-gene
interaction matrix of the dataset by maximizing the entropy of the
genomic system (see Methods). To maximally reflect the gene-gene
interaction information of the system through a 2D spatial configura-
tion of genes, we transform the dataset into a series of cell-specific
genomaps by optimizing a transport function. As the possible ways of
gene placement into a 2D grid for a cell is a factorial of the number of
involved genes, a robust optimization of the transport function is
imperative to reliably construct a genomap. In general, a genomap
possesses the basic characteristics of an image with the pixelated
configuration manifesting the gene-gene interactions (Fig. 1a) and
provides a comprehensive representation of the gene expression data.
After the construction of the genomaps, we extract the configurational
features of the genomic interactions by using an efficient convolu-
tional neural network (CNN) named genoNet (see Fig. 1b and Meth-
ods). In this way, deep correlative features of the genes are extracted
effectively from the data for subsequent decision-making. We show
that, for a wide variety of applications, including cell clustering and
recognition, gene signature extraction, single cell data integration,
cellular trajectory analysis, dimensionality reduction, and visualiza-
tion, the proposed approach substantially outperforms the state-of-
the-art methods. The proposed technique presents a unique paradigm
for analyzing genomic data or other forms of tabular data and pro-
mises to broadly impact data science.

## Results

### Genomap enables extraction of configurational features of gene-gene interactions for highly accurate recognition of cell types

The potential of genomap is first illustrated by using Tabula Muris
(TM) scRNA-seq dataset (transcriptomics of 20 mouse organs, con-
taining 55 different cell classes listed in Supplementary Fig. S3)[22]. The
gene expression data of 54,865 cells, each with 19,791 genes, were
analyzed. In Fig. 2, we show genomaps of 100 cells belonging to 10
representative classes (genomaps of the remaining 45 classes are
presented in Supplementary Fig. S2). Note that the first and last five
rows in Fig. 2 correspond to normal and stem cells, respectively, and
their genomaps appear very differently. Within each category, the
genomap pattern also varies from class to class. For example, for
keratinocyte cells (1st row), the genes with high expressions are posi-
tioned in a doughnut-shaped circular region close to the boundary of
the map. Interestingly, the high expression genes of another skin cell,
the epidermal basal cell (2nd row), also appears in a doughnut shape.
However, the expressions of the genes inside the doughnut are higher.
For another example, in immature B-cells (last row), the genes with
high expressions show up in the central region, whereas the genes with
low expressions occupy the entire remaining space. It is worth noting
that the genomap of the epidermal basal cell (2nd row) is similar to that
of the bladder urothelial cell (3rd row). Although these two types of
cells are from different parts of the body (skin and bladder), they both
belong to the category of basal cell and share similar biological char-
acteristics. Another interesting similarity is observed between the
genomaps of granulocytopoietic and promonocyte cells, which
represent different branches of stem cell development when the GM-
CSF enzyme is present. On the other hand, the genomaps appear

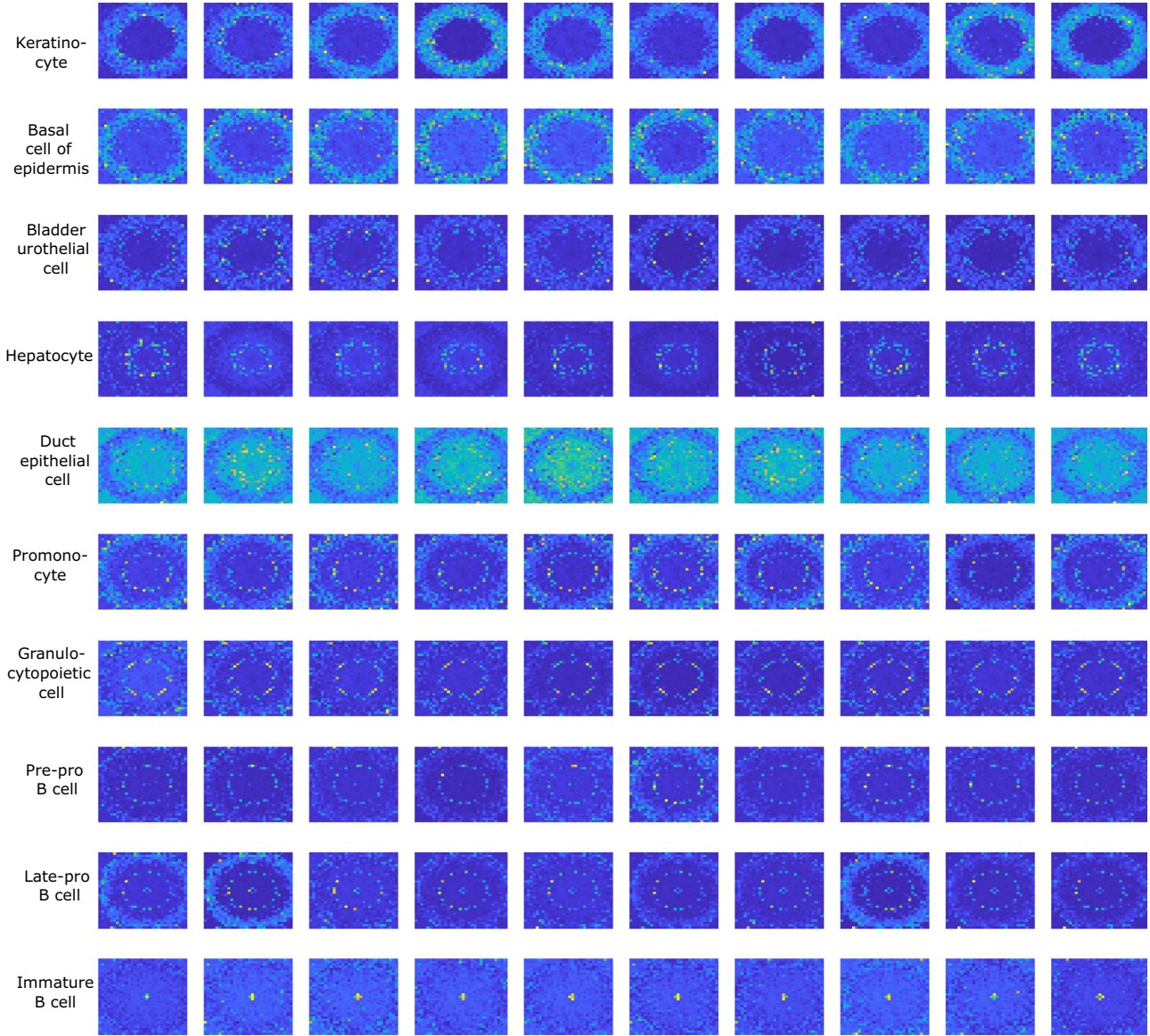

**Fig. 2 | Genomaps of 100 cells belonging to 10 different classes from Tabula Muris dataset.** Each row in the figure corresponds to a class. For each class, the 10 cells show very similar patterns of genomap. Here, the smallest value in genomaps is denoted by blue and the largest value is denoted by yellow.

differently for cells of different functions, such as hepatocytes from the liver (4th row) and duct epithelial cells from the breast (5th row).

Genomap also provides insights into the cellular developmental process. For example, B cells undergo 6 stages as they develop: lymphoid progenitor cells, early pro-B cells, late pro-B cells, large pre-b cells, small pre-B cells, and finally immature B cells. From the genomaps of pre-pro B-cells (8th row), late-pro B cells (9th row), and immature B cells (last row) in Fig. 2, we observe that the genes on the two concentric circles are highly expressed in both pre-pro and late-pro B cells. However, the gene expression levels at the center are very low in the former case, but increase to higher values in the late-pro B cells. At the last stage (i.e., immature B-cells), the genes at the center exhibit very high expression values while the two concentric rings fade away.

It is intriguing that the difference in the genomaps of many cell types is so obvious that they can be identified visually even without deep learning. Our genoNet (see Methods) analyses of the genomaps are presented in the supplementary along with the results obtained using ten representative existing methods (Supplementary Fig. S4). The classification accuracy of our approach reaches 93%, which is 6%,

and 21% higher than that of Cell-ID[23] and SingleR[24] methods, respectively.

We now showcase our approach by analyzing the ischemic sensitivity dataset, a very large dataset that is part of the human cell atlas[25] (see Methods for data description). In this dataset, the sequenced cells are from three important human organs - lung, esophagus and spleen. The genomaps of the cells from the lung are shown in Fig. 3. Again, different genomap patterns are observed for different classes of cells. The uniform manifold approximation and projection (UMAP)[26] visualizations of the raw data for the three organs are shown in Fig. 4(a), where it is seen that the data classes are not well-separable. However, with the use of genomaps and genoNet, the data classes are better separated at the fully connected layer (Fig. 4(b)) for both training and testing data. The classification results of different methods are shown in Fig. 4(c). Remarkable accuracies of 88%, 96% and 85% are achieved by our approach for the lung, esophagus and spleen datasets, respectively, which are at least 5% higher than the best performer among the existing techniques. Furthermore, our approach shows sustained superior performance even with reduced size of training datasets. To illustrate this, we analyze the performance of different

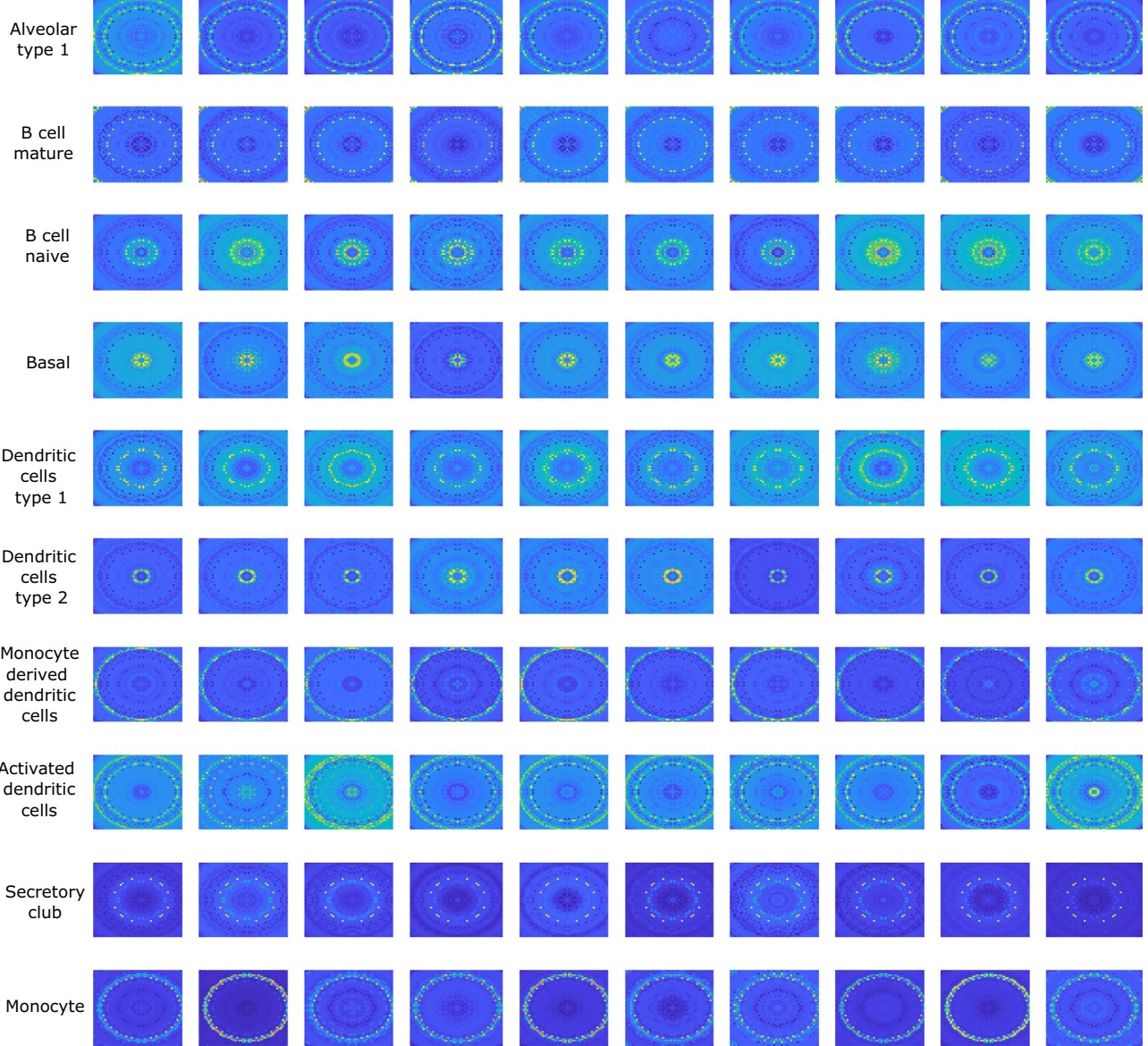

**Fig. 3 | Genomaps of 100 cells belonging to 10 different classes from ischaemic sensitivity dataset acquired from the lung.** Each row in the figure corresponds to a class. For each class, the 10 cells show very similar patterns of genomap. As examples, in mature B cells (2nd row), the genes in a circular ring close to the boundary have high expressions whereas the genes in other areas have low expressions. On the other hand, in naive B-cells (3rd row), the genes in a circular region close to the center have very high expressions. Here, the lowest gene expression value is denoted by blue and the highest value is denoted by yellow.

techniques when only 30% and 50% of the total data are used for training. The classification results shown in Supplementary Fig. S7 clearly demonstrate the superiority of the proposed approach over the existing methods. Noteworthy, the performance of our technique drops only by 2–3% for the study with 30% of total data for training, whereas the decrease in classification accuracy of all other techniques (except ACTINN[27] and Vec2image[28]) is at least 7%.

A scRNA-seq dataset profiled from mouse T cells[29] is used to further demonstrate the success of the proposed approach. Elyahu et al. used scRNA-seq and multidimensional protein analyses to profile thousands of CD4 T cells obtained from young and old mice. The UMAP visualization of the raw data is shown in Fig. 5a, in which the data classes are not well-clustered. We converted the data to genomaps and our genoNet analysis shows that all seven classes are now well separated at the fully connected layer (Fig. 5a). The classification accuracy of our approach is 87%, which is at least 17% higher than the state-of-the-art Cell-ID technique, 10% higher than SingleR, and 5% higher than ACTINN and Vec2image (Fig. 5b). To support the notion that genomap

is the key driver for high-performance classification, results of genomap+ACTINN and 1D expression+genoNet are also included in Supplementary Fig. S5.

**Genomap-based analysis outperforms the state-of-the-art technique in discovering cell-specific and class-specific gene sets**
The proposed cartography approach is able to accurately find gene sets characterizing specific cells and cell types. To achieve this, we compute the class-activation map by using GRAD-CAM[30] for the genomaps of TM dataset shown in Fig. 2. The class-activation maps are shown in Fig. 6, in which the genes that are most important to the genoNet decision-making have higher values. The findings here are validated by using an independent experimental technique called sci-ATAC-seq, which measures the gene activity by quantifying the chromatin accessibility in a single cell[31]. We compute the average gene activity scores from sci-ATAC-seq dataset in five types of cells, which are common between the TM dataset and sci-ATAC-seq dataset. The gene activity scores for the marker genes of B cells, T cells

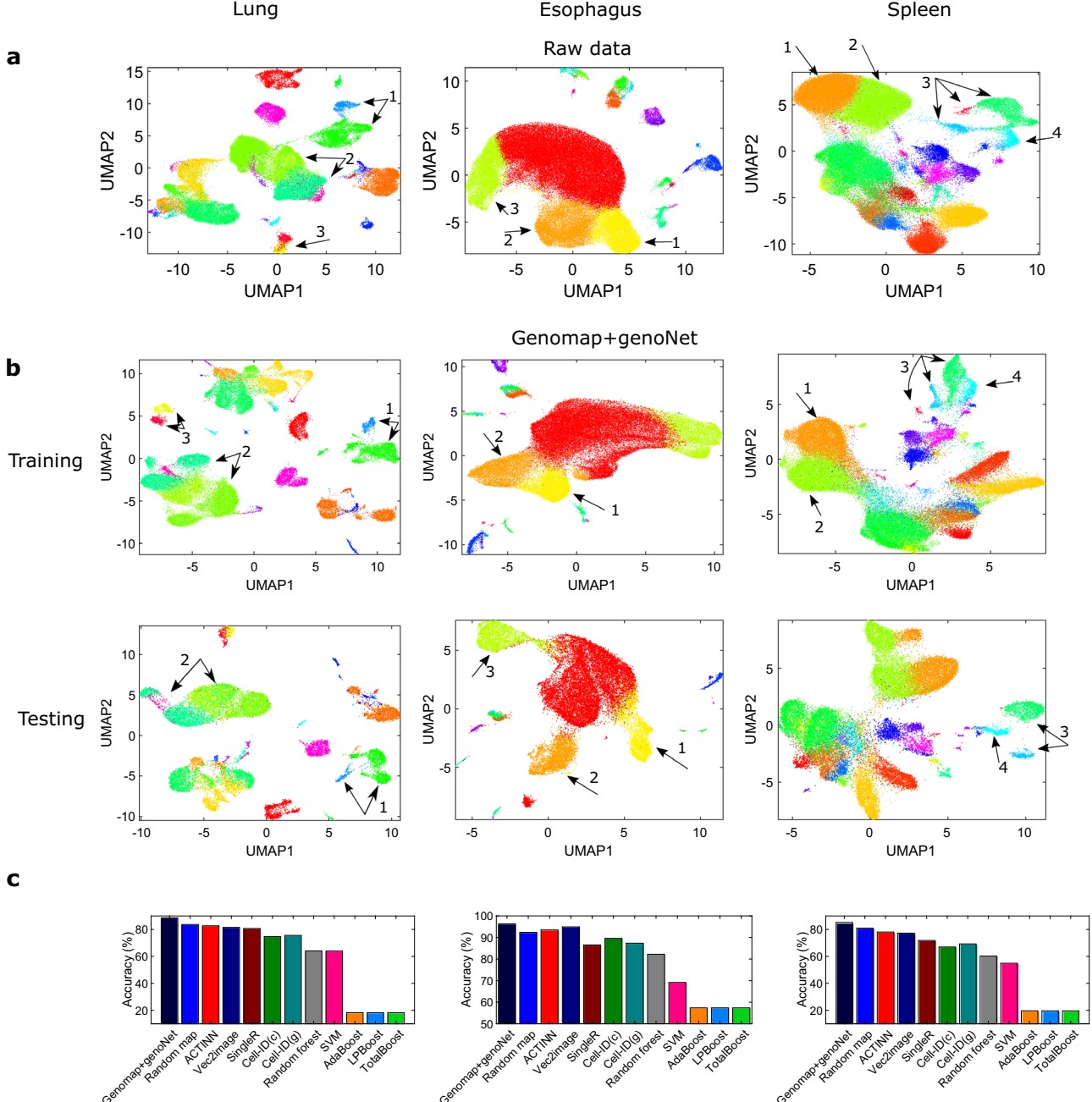

**Fig. 4 | Visualization of ischaemic sensitivity dataset (left-lung, middle-eso-phagus, right-spleen). a** UMAP visualizations of raw data. **b** UMAP visualizations of the genomap features at the fully connected layer of the genoNet. Major improvements in cluster separation are indicated by arrows. Color legends of the data classes are added in Supplementary Fig. S22. **c** Classification accuracy of different techniques including genomap+genoNet. Here, Cell-ID(c) and Cell-ID(g) denote Cell-ID technique with cell-to-cell and cell-to-group matching formulation. Source data are provided as a Source Data file.

and hematopoietic (HP) cells namely *CD3D*, *CD3E*, *CD79A*, *CD79B*, and *CD34* in sci-ATAC-seq data are shown in Supplementary Fig. S8 (a-left). For benchmarking, the importance scores of the genes in different cells are also computed by the Cell-ID technique[23], and the results are shown in (a-right) along with the genomap findings in (b-left). It is seen that both Cell-ID and our approach show higher activities of *CD3D* and *CD3E* genes in T cells and *CD79A* and *CD79B* genes in B cells. This is reasonable as *CD3D*, *CD3E* and *CD79A*, *CD79B* are established markers for T cells and B cells, respectively[31]. However, our approach shows much lower activities of these genes in other cells such as Monocyte, NK cell and HP cells, which aligns well with the ground truth sci-ATAC-seq data. Again, our approach outperforms Cell-ID when their performances are compared with the

ground truth sci-ATAC-seq data in terms of cosine similarity (Supplementary Fig. S8 (b-right)).

**Genomap affords a fine-tuning framework for accurate integration of scRNA-seq data acquired with different technologies**

Single cell data integration, which denotes the process of projecting gene expression data from different sequencing technologies or protocols to a common subspace, aims to mitigate measurement specific bias so that all the data from different measurements can be utilized synergistically for downstream inferences[32–34]. With the integration, for example, a model trained on one set of scRNA-seq data can be applied to analyze another dataset(s) obtained under different condition(s). The decision-making process of our approach based on the Genomap

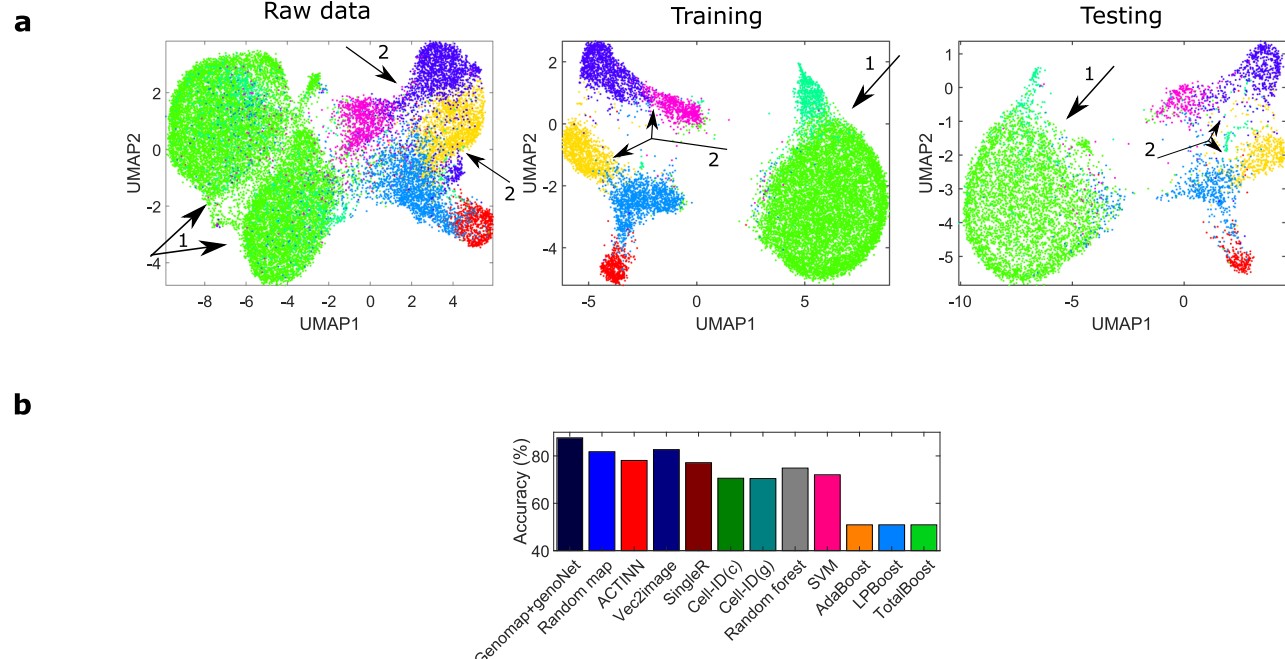

**Fig. 5 | Analysis of T cell landscape dataset. (a** left) UMAP visualizations of the raw data. (**a** middle and right) UMAP visualizations of features from the fully connected layer of the genoNet for the training and testing datasets. Major improvements in cluster separation are indicated by arrows. Color legends of the data classes are added in Supplementary Fig. S23. **b** Classification accuracies of the genomaps with the proposed approach and existing techniques. Source data are provided as a Source Data file.

construction and genoNet analysis depends primarily on the interactions among the genes, which is less susceptible to the experimental protocols and provides an efficient framework for single cell data integration. To demonstrate this advantage, we choose five human pancreatic datasets from the works of Segerstolpe et al.[35], Baron et al.[36], Muraro et al.[37], Xin et al.[38], and Wang et al.[39]. The datasets were acquired using SMART- Seq2, inDrop, CEL-Seq2, and SMARTer scRNA-seq technologies, respectively. After pre-aligning the datasets by canonical correlation analysis (CCA) from Seurat[32], the gene expressions are converted into five sets of genomaps. A genoNet model is then trained by using four sets of genomaps and the model is applied to classify the fifth set of genomaps. The results are shown in Fig. 7a–c along with the state-of-the-art techniques: Seurat v3[32], Harmony[33], and online iNMF[34]. From the UMAP visualizations of the results colored by the batch type (Fig. 7a), it is seen that Seurat, Harmony and online iNMF integrate the datasets to some extent. In our integration, most of the data are well-mixed and the effect of batch type is greatly reduced on the UMAP visualization. The same visualizations colored by the actual cell classes are shown in Fig. 7b. Our approach finds most of the 15 clusters in the dataset, with better cluster quality than other techniques. From the label transfer efficiency barplot shown in Fig. 7c, it is seen that our model leads to an accuracy of 97% for Segerstolpe dataset, whereas other methods can at best reach an accuracy of 89%. Improved performance is also obtained for other combinations of the datasets (see the last two columns of Fig. 7c and Supplementary Fig. S10).

### Genomap enables cellular trajectory mapping with highest fidelity
We select a scRNA-seq dataset containing cells at different stages throughout embryogenesis of a proto-vertebrate (sea squirt)[40]. In total, 90,579 cells from 10 different development stages were sequenced. The t-distributed stochastic neighbor embedding (t-SNE)[41], UMAP, and PHATE[14] visualizations of the raw data are shown in Fig. 8a. It is seen that in the t-SNE and UMAP visualizations, the data are clustered and show no cell trajectories. In unsupervised PHATE visualization, while some branchings of the cell development are observed, the transitions of different stages are not clearly reflected in these figures. No trajectory can be seen in the results of supervised PHATE[42]. We created the genomaps for the cells and trained the unsupervised genoNet model (see Methods). The PHATE visualization of genoNet features at the final fully connected layer of the genoNet model is displayed in (a-2nd row last column). Continuous trajectories are seen for different branches. The transitions of cellular stages at different time points are more distinct in comparison to the competing techniques. Quantitative comparisons between the existing PHATE and our unsupervised approach are shown in (b) in terms of DEMaP[14] index. A higher DEMaP index for the proposed approach indicates that the genomap embedding preserves the characteristics of HD data better than other techniques.

### Genomap offers accurate unsupervised dimensionality reduction, visualization, and clustering of HD data
The superiority of the proposed approach in dimensionality reduction, visualization, and clustering analysis is demonstrated by using a scRNA-seq dataset profiled from mouse retinal bipolar cells (BCs)[43]. From a population of 27,499 BCs, Shekhar et al. identified 15 types, including two new types of neurons, by leveraging the class specific gene signatures. Our analytic tool can also be utilized to find the data classes. For this purpose, we converted the data from Shekhar et al. to genomaps and applied unsupervised genoNet to reduce the data dimension and extract underlying features for the above tasks. The t-SNE, linear discriminant analysis (LDA)[44], Siamese network[45], and supervised and unsupervised UMAP visualizations of the raw data are shown in Fig. 9a. It is seen that several data classes such as BC3B and BC4 are inseparable. The t-SNE and UMAP visualizations on the data embeddings from the fully connected layer of the genoNet are shown in (b), where it is seen clearly that most of the data classes become well-separated (major improvements in cluster separation are indicated by arrows). The quantitative comparison among different techniques in terms of clustering accuracy, adjusted Rand (AR), Silhouette, and normalized mutual information (NMI) indices are shown in (c). Louvain

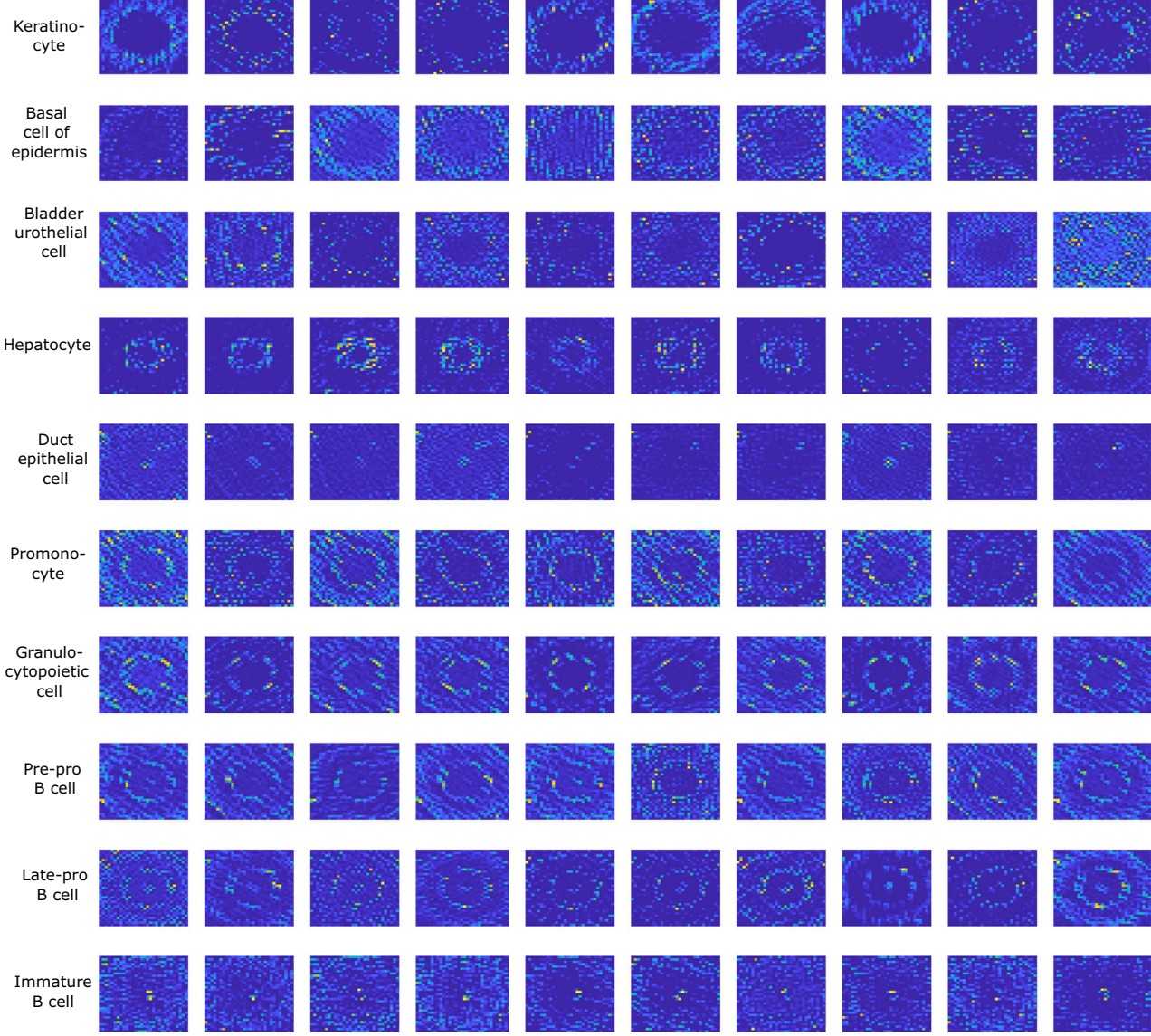

**Fig. 6 | Class activation maps of the cells displayed in Fig. 2.** The intensity of a pixel in the map denotes the importance (scaled from 0 to 1) of the corresponding gene. For each class, very similar patterns of class activation maps are observed for the 10 cells, confirming the existence of a number of genes specific to a data class (class-specific gene set). Here, the lowest and highest values in the maps are denoted by blue and yellow colors, respectively.

technique[46,47] was employed for clustering the embeddings from different methods. It is seen that the visualizations from the genoNet embeddings achieve much higher cluster quality scores than the competing methods.

## Discussion

How to discover patterns and gain insights from HD data with ever-increasing size, dimensionality and complexity represents a common theme in modern data science, especially in genomics and biomedicine. Traditionally, these data are expressed in a vector or matrix form and processed by direct embedding them into a low dimension through either an analytical (e.g., principal component analysis (PCA)[48], *t*-SNE, UMAP, and kernel PCA) or data-driven (e.g., variational autoencoder (VAE)[49], feature-augmented embedding machine (FEM)[50], and Siamese network[45]) approach. Unfortunately, none of these approaches explicitly takes the underlying discriminant biological characteristics of the dataset into consideration. Rather than amending the traditional techniques, here we take a radically different route to transform the data into a configured format, which enables us to extract the configuration features of the genomic interaction for

various applications. The proposed cartography technique overcomes the fundamental limitations of the traditional approaches that have frustrated efforts to extract the most discriminative information in the system. Our results show that the configurational features of the genomaps provide valuable contextual cues for us to gain a better understanding of the genomic system and build predictive models with dramatically improved performance[51–56]. With a synergistic use of deep learning, the proposed approach shows significant potential for high performance processing of HD data. Given the exquisite insights provided by the genomap and its remarkable performance, the technique should be valuable for uncovering new and potentially unexpected biological discoveries in the future.

How to maximize the utility of a given set of data for robust decision-making has been the holy grail of big data research[57–59]. In an image, single or multiple concepts (i.e., high-level collective information) can be conveyed by a group of adjacent pixels. In contrast, a group of data values in a table usually does not possess any collective information by default. As shown in this work, tabulated data can often be better represented in image format with the data interactions casted into the pixelated configuration. Genomap bridges the gap

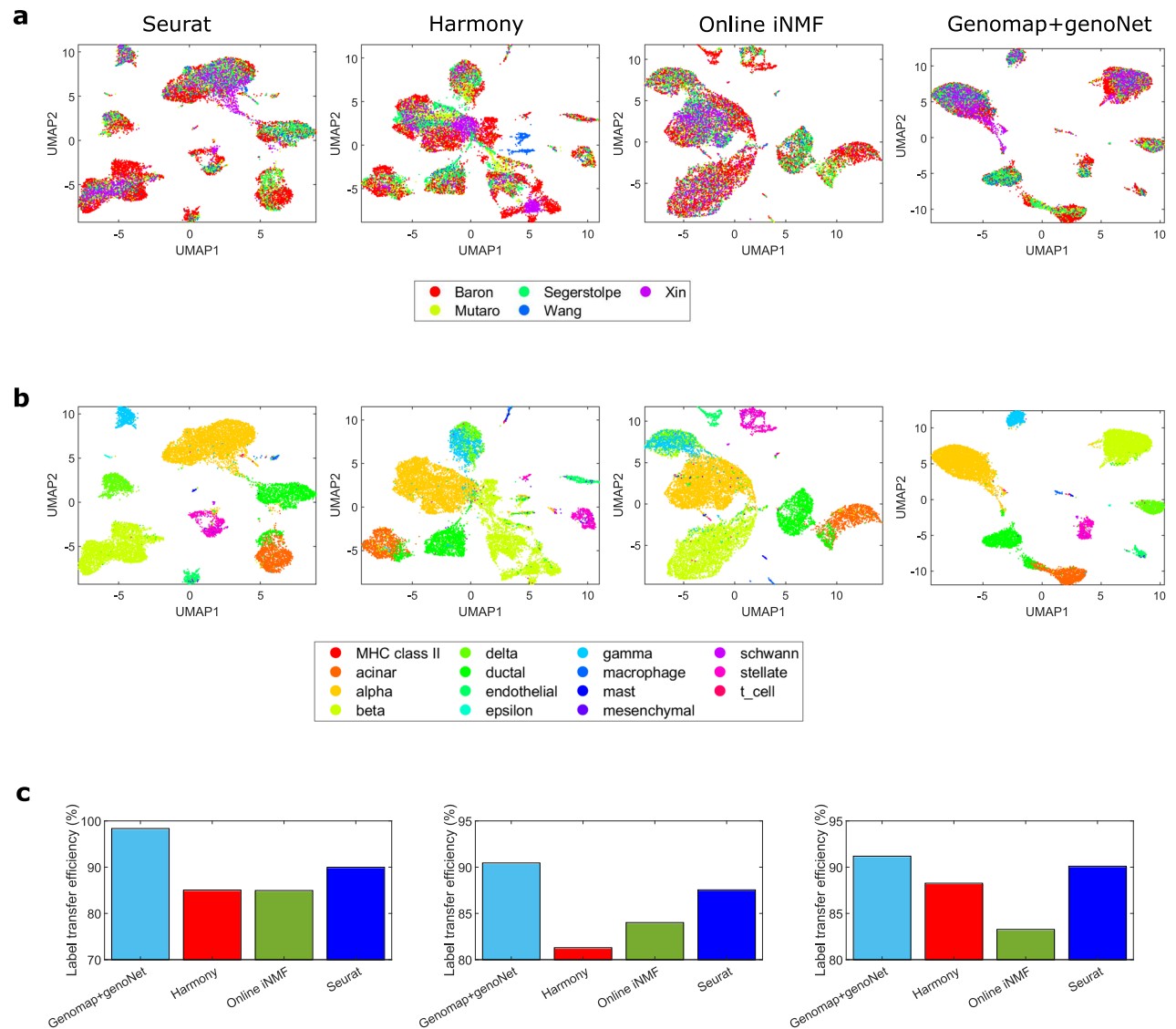

**Fig. 7 | Integration of the single cell datasets obtained by using five different measurement protocols.** In (**a**) UMAP visualizations of embeddings resulted from different integration techniques (Seurat, Haromony, Online iNMF, and genomap +genoNet) are shown. Data of different measurement protocols are denoted by different colors. In (**b**), the same UMAP visualizations of the embeddings are shown with the cell classes denoted by different colors. (**c**) Label transfer accuracy of different integration techniques (left to right-Segerstolpe, Baron, and Muraro datasets). Source data are provided as a Source Data file.

between tabular and image datasets and allows us to take advantage of existing image processing toolkits for better data analysis. In reality, there are numerous ways to convert a set of genomic data to 2D maps. The gist of genomap is to lay out the genes in such a way that the interactive information among the genes are maximally conveyed. In general, genes interact with each other in complex fashions and can be characterized by a gene-gene interaction network in high dimension[18,19]. To a large extent, the genomap represents an optimal projection of the interactions to a 2D space. It should be noted that while genomaps can also be created in other dimensions (such as 3D as demonstrated in Supplementary Fig. S6), previous studies show that the ratio of performance to computational effort is better with a 2D CNN model[60,61].

The proposed cartography approach provides a unique computational paradigm of high performance genomic data analysis. The potential of the strategy is demonstrated by a broad spectrum of applications, including cell recognition, single cell data integration, cellular trajectory analysis, gene signature extraction, dimensionality reduction, clustering and visualization tasks. However, the technique can also be applied to many other important problems. Furthermore,

the concept of reconfiguring tabulated data into an image is quite general and we envision that the technique will find valuable applications in dealing with HD data across disciplines, such as healthcare, finance, and physical sciences. We note that there were a few attempts in the literature to convert gene expression data into images for deep learning[28,62–64]. In these methods, however, an image is usually made heuristically by projecting the HD data onto a 2D plane without any explicit constraint(s) on the spatial locations of the genes. As an example, In vec2image[28], *t*-SNE (or other) embedding method is used to create the images. As a result, similar genes get clustered under the assumption of t-distribution without explicit constraints on their spatial positions for maximizing entropy. Therefore, many genes located in the outer region of the clusters have fewer neighbors than those located in the center of the cluster. The same is true for those non-clustered (i.e., isolated) genes. The reduction of the number of neighbors to many genes deteriorates the information extraction efficiency of the CNNs (because of the limited receptive field of CNN). In contrast, the genomap here is established on a solid theoretical foundation with the goal of finding the optimal spatial configurational representation of the gene-gene interactions of the system. Thus, as

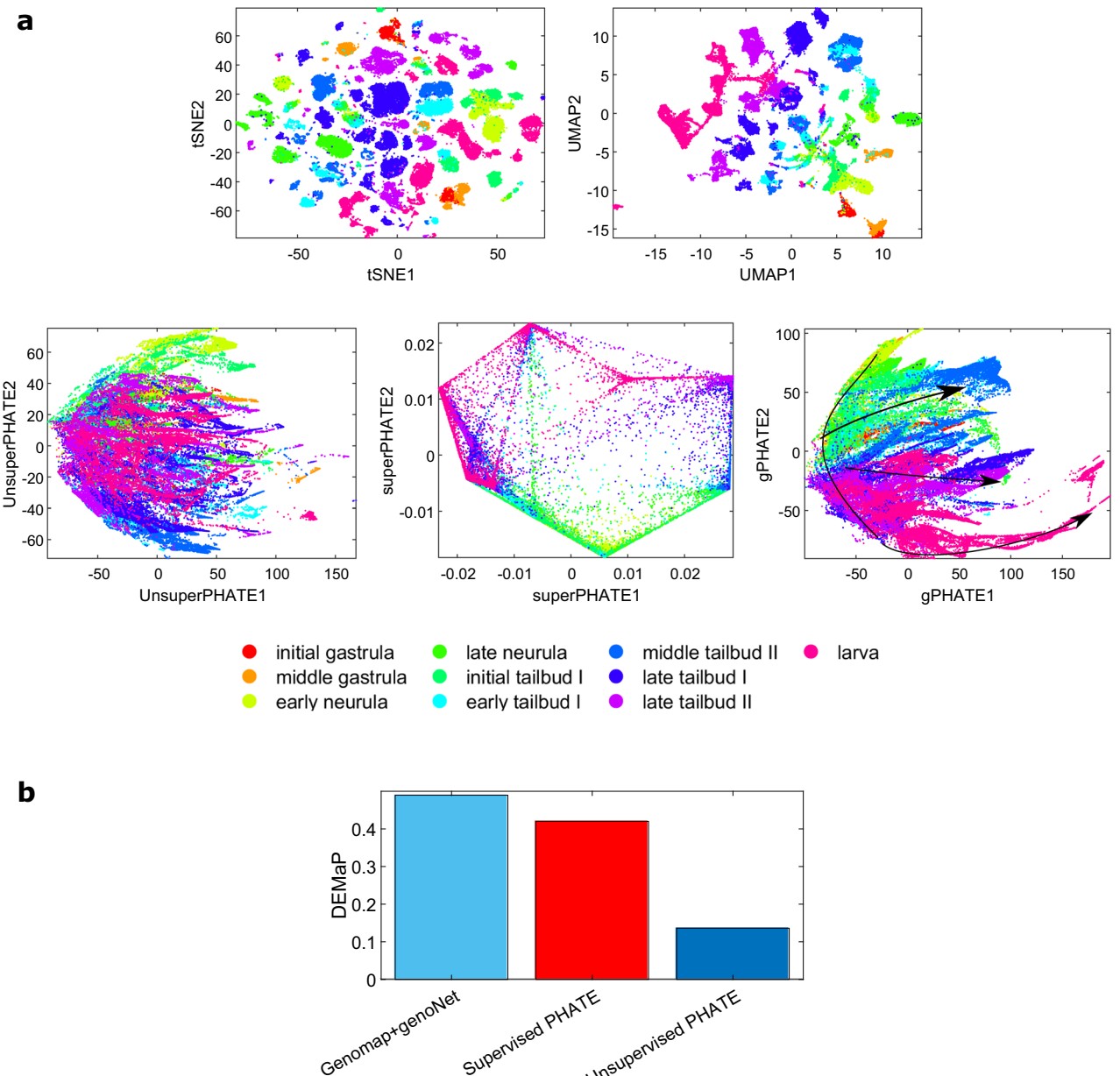

**Fig. 8 | Cellular trajectory analysis of proto-vertebrate dataset by using different techniques.** (**a**-1st row) *t*-SNE and UMAP visualizations of the data. (**a**-2nd row, first and second columns) Unsupervised and supervised PHATE visualizations of the data. (**a**-2nd row, last column) PHATE visualization of the embedding from the unsupervised genoNet. In contrast to the results of existing techniques, the transitions of cells from initial grastula to larva are quite evident in the visualization of the proposed approach. (**b**) DEMaP computed from embeddings of different techniques. Source data are provided as a Source Data file.

shown in a variety of applications in the Results section, the deep analysis of genomaps leads to discoveries of highly distinctive patterns of complex genomic data and provides a potentially useful analytic technique. Theoretical analyses of the information content of genomaps and images from other methods are presented in supplementary section 2 (Supplementary Fig. S1 and Table S1).

We compared the performance of the proposed approach with different state-of-the-art techniques in different applications. In all our studies, we used 70% of the data for training and the rest for testing, which is a standard practice in machine learning. It should be emphasized that the advantage of our approach over existing ones is even more pronounced in the case of smaller datasets (supplementary Fig. S7). On another note, as exemplified in Supplementary Figs. S11–S16, where the number of cells as a function of the involved class is plotted, most biological datasets are highly imbalanced. The

superior performance of our approach as demonstrated in the Results for these difficult classification tasks is of particular importance and shows its potential for practical applications. Note that, for computational efficiency, in all our studies, we used genoNet with a simple architecture (see Supplementary Table S2 and Methods). Generally, the performance of genoNet can be further improved by adding more convolutional layers. For example, for the TM dataset, genoNet with 2 and 3 convolutional layers improves the classification accuracy by 0.6% and 0.9%, respectively. More complex CNNs (such as Xception[65], GooLeNet[66], and DenseNet[67]) would generally provide comparable or slightly improved performance due to deeper feature extraction (see Supplementary Fig. S9 for an example of Xception analysis of the TM data), but at the cost of dramatically increased computational burden. For example, for the TM dataset, genoNet with a single convolutional layer takes 148 seconds to finish training 150 epochs on a computer

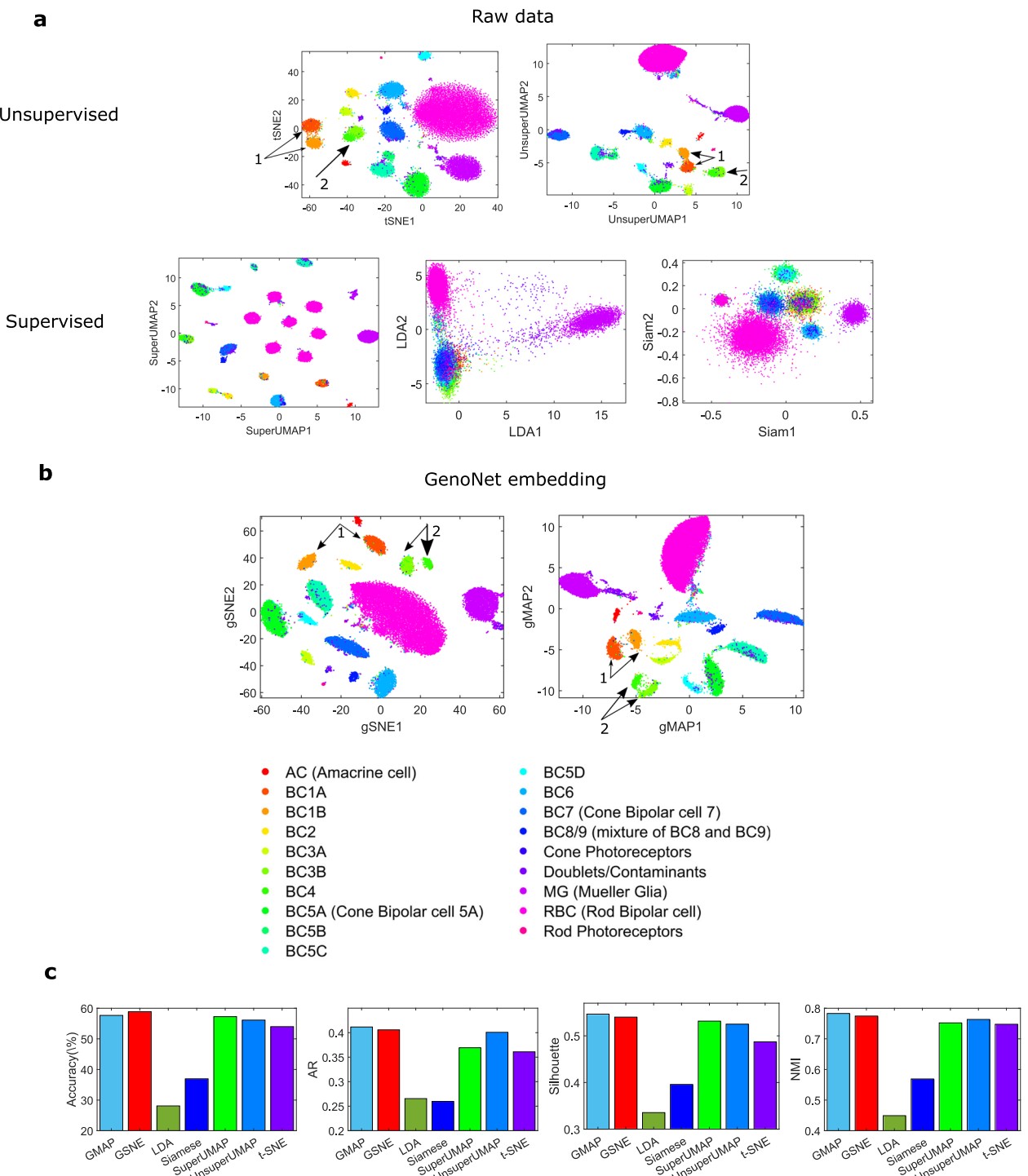

**Fig. 9 | Dimensionality reduction, clustering, and visualization of the retinal bipolar cells by different techniques. a** *t*-SNE, LDA, Simaese network, and unsupervised and supervised UMAP visualizations of the raw data. **b** *t*-SNE and UMAP visualizations of the embeddings from unsupervised genoNet. Major improvements in cluster separation are indicated by arrows. **c** Quantitative comparison of cluster quality of embeddings from different techniques in terms of accuracy, AR, Silhouette and NMI indices (from left to right panels). The bar heights denote the mean values of the indices for for 1000 different initializations of Louvain clustering. GSNE and GMAP denote the *t*-SNE and UMAP embedding obtained from genoNet features. Source data are provided as a Source Data file.

with Intel Core i9 CPU, 128GB RAM, and NVIDIA RTX A5000 GPU, whereas Xception, GoogLeNet, and DenseNet require more than 8 hours for the same task. It should be noted that the computational overhead for genomap calculation is rather small. For example, a dataset containing 10,000 cells and 1000 genes is processed in only 61 seconds on the aforementioned computer.

We note that, in the construction of genomaps, only two-way gene-gene interactions are considered[68]. However, the probability of higher order interactions is generally much lower and this issue is discussed in detail in section 1 of the supplementary. In creating the genomaps, following the common practices in single-cell RNA-seq analysis[69], we selected 5–10% highly variable genes from the datasets. However, we

**Table 1 | List of notation**

| Symbol | Explanation | Symbol | Explanation |
|---|---|---|---|
| $H$ | Entropy | $X$ | Random variable |
| $m$ | Number of cells | $n$ | Number of genes |
| $p(x) = P(X = x)$ | Probability mass function of $X$ | $\boldsymbol{\Omega}$ | Covariance matrix |
| $\rho$ | Interaction strength between genes | GW | Gromov-Wasserstein discrepancy |
| $\mathbf{C}$ | Scaled interaction matrix | $\bar{\mathbf{C}}$ | Distance matrix of 2D grid points |
| $\mathbf{T}$ | Coupling matrix | $L$ | Loss function |
| $\mathbf{x}_i$ | Expression level of i-th gene | $\mathcal{L}$ | 4-way tensor of the loss function |
| $\mathbf{u}$ | Relative importance of genes | $\mathbf{v}$ | Relative importance of grid locations |
| $p$ | Number of rows in 2D grid | $q$ | Number of columns in 2D grid |
| $\top$ | Transpose operation | $\mathcal{E}$ | Optimal transport function |
| $\epsilon$ | Regularization parameter | $\tau$ | Step size in Sinkhorn iteration |
| $i, j, k, l$ | Index variables | $f_1, f_2, h_1, h_2$ | Functions |
| $\mathbf{w}^*$ | Optimal regression coefficient vector | $\mathbf{e}$ | Residual |

note that genomap performance becomes better when more genes are included in the analysis (see Supplementary Table S4). For a very large number of genes (such as more than 10000), the genomaps may be very sparse and the performance of the proposed approach may be compromised because of the inefficiency of genoNet to capture the relationship between different genes. However, in such cases, PCA or other dimensionality reduction methods (such as non-negative matrix factorization (NNMF)[70], multi dimensional scaling (MDS)[71], and FEM[50]) can be applied on the dataset and genomaps can be created from the principal components (or components from other methods). Results for one such case of genomaps created from different numbers of principal components computed from the comprehensive classification of retinal dataset are reported in Supplementary Table S5.

Gene expression profiling powered by advanced data analytic methods is becoming increasingly essential in biomedical research. In this work, we have presented a systematic strategy for deep feature extraction from the deluge of genomic data. We have shown that the integration of sophisticated gene-gene interactions into data processing pipelines presents a unique opportunity to radically improve analytic performance. With the construction of genomaps and the effective use of deep learning, the proposed approach shows significant potential in discovering unique data patterns and gaining useful insights into various biological systems. Finally, we emphasize that the proposed cartography approach is broadly applicable to deal with HD data involving not only gene expressions but also tabular data across different disciplines.

## Methods
### Overview
To reconfigure the gene expression vector of a cell into a 2D genomap, we first obtain the pairwise interaction strength matrix that maximizes the entropy of the data. We then place the genes in a 2D grid in such a way that their pairwise interaction is preserved maximally. We utilize the optimal transport optimization (minimization of Gromov-Wasserstein discrepancy between interaction-space of genes and Euclidean space of 2D grid) to solve the problem efficiently.

### Theory
Let us assume that we have a dataset, $\mathbf{x} \in \mathbb{R}^{m \times n}$ from an experiment on $m$ number of cells (each cell has $n$ number of genes). Our objective is to restructure the $n$ genes of each cell into a 2D grid of size $p \times q$, $p \times q \geq n$ to maximize the entropy of the data.

**Entropy.** Entropy is frequently used in information theory to measure the information content of a system. Mathematically, entropy measures the uncertainty associated with a random variable or a random

system[72]. The entropy $H(X)$ (see Table 1 for details of the notations) of a discrete random variable $X$ can be written as

$$H(X) = -\sum_{x \in \mathcal{X}} p(x) \log p(x), \tag{1}$$

where $p(x) = P(X = x)$, $x \in \mathcal{X}$, denotes the probability mass function (pmf) of the random variable $X$, and $\mathcal{X}$ is a finite set (such as $\{1, 2, ..., \}$)[73]. The entropy has been used for analyzing the interaction among attributes in biological and other datasets in the literature[74–76].

**Pairwise interaction of genes.** The expression data of the $i$-th gene in $m$ cells can be written as a vector $\mathbf{x}_i = (\mathbf{x}_i^1, ..., \mathbf{x}_i^m)$. For $n$ number of genes, we have $n$ vectors $\mathbf{x}_1, ..., \mathbf{x}_n$. Our goal is to restructure the genes in each cell in such a way that maximizes the entropy of the gene expression vectors[68]

$$H = -\sum_{\mathbf{x}} p(\mathbf{x}) \ln p(\mathbf{x})$$

subject to the constraints

$$\sum_{\mathbf{x}} p(\mathbf{x}) = 1.$$

The probability mass function for the gene expressions, which maximizes the system entropy is given by a multivariate Gaussian distribution parametrized by the mean $\langle \mathbf{x} \rangle$ and the covariance matrix $\boldsymbol{\Omega}$ as follows[77]:

$$p(\mathbf{x}; \langle \mathbf{x} \rangle, \boldsymbol{\Omega}) = (2\pi)^{-n/2} \det(\boldsymbol{\Omega})^{-1/2} \exp\left(-\frac{1}{2}(\mathbf{x} - \langle \mathbf{x} \rangle)^{\top} \boldsymbol{\Omega}^{-1}(\mathbf{x} - \langle \mathbf{x} \rangle)\right).$$

Here, the covariance matrix is defined as:

$$\boldsymbol{\Omega}_{ij} = \langle \mathbf{x}_i \mathbf{x}_j \rangle - \langle \mathbf{x}_i \rangle \langle \mathbf{x}_j \rangle, \tag{2}$$

where

$$\langle \mathbf{x}_i \rangle = \frac{1}{m} \sum_{k=1}^{m} \mathbf{x}_i^k,$$

and

$$\langle \mathbf{x}_i \mathbf{x}_j \rangle = \frac{1}{m} \sum_{k=1}^{m} \mathbf{x}_i^k \mathbf{x}_j^k.$$

The pairwise interaction strength between $\mathbf{x}_i$ and $\mathbf{x}_j$ can now be computed from the covariance matrix as follows[77]:

$$\boldsymbol{\rho}_{ij} = \begin{cases} -\dfrac{(\boldsymbol{\Omega}^{-1})_{ij}}{\sqrt{(\boldsymbol{\Omega}^{-1})_{ii}(\boldsymbol{\Omega}^{-1})_{jj}}} & \text{if } i \neq j, \\ 1 & \text{if } i = j. \end{cases} \quad (3)$$

**Genomap construction.** The problem of constructing genomaps, i.e., optimally placing $n$-genes to $n$ positions of the 2D grid of $p \times q$ ($n \leq p \times q$) can be written as Gromov-Wasserstein discrepancy between the scaled pairwise interaction strength matrix $\mathbf{C}$ of $n$ genes and the distance matrix ($\bar{\mathbf{C}}$) of the 2D grid space. Both the matrices $\mathbf{C}$ and $\bar{\mathbf{C}}$ are of size $n \times n$. We define the Gromov-Wasserstein discrepancy between matrices $\mathbf{C}$ and $\bar{\mathbf{C}}$ as follows[78]:

$$\text{GW}(\mathbf{C},\bar{\mathbf{C}},\mathbf{u},\mathbf{v}) \stackrel{\text{def.}}{=} \min_{\mathbf{T} \in \mathcal{C}_{\mathbf{u},\mathbf{v}}} \mathcal{E}_{\mathbf{C},\bar{\mathbf{C}}}(\mathbf{T}), \text{ where } \mathcal{E}_{\mathbf{C},\bar{\mathbf{C}}}(\mathbf{T}) \stackrel{\text{def.}}{=} \sum_{i,j,k,\ell} L\left(\mathbf{C}_{i,k},\bar{\mathbf{C}}_{j,\ell}\right) \mathbf{T}_{i,j} \mathbf{T}_{k,\ell}. \quad (4)$$

Here, the matrix $\mathbf{T}$ is a coupling between the two spaces on which $\mathbf{C}$ and $\bar{\mathbf{C}}$ are defined, $\mathbf{u}$ and $\mathbf{v}$ are vectors containing relative importance of the genes and the locations in the genomap. $L$ here is a loss function to account for the discrepancy between the matrices and defined as the Kullback-Leibler divergence $L(a,b) = \text{KL}(a|b) \stackrel{\text{def.}}{=} a \log(a/b) - a + b$. Introducing the 4-way tensor

$$\mathcal{L}(\mathbf{C},\bar{\mathbf{C}}) \stackrel{\text{def.}}{=} \left( L\left(\mathbf{C}_{i,k},\bar{\mathbf{C}}_{j,\ell}\right) \right)_{i,j,k,\ell},$$

we have

$$\mathcal{E}_{\mathbf{C},\bar{\mathbf{C}}}(\mathbf{T}) = \langle \mathcal{L}(\mathbf{C},\bar{\mathbf{C}}) \otimes \mathbf{T},\mathbf{T} \rangle.$$

Here $\otimes$ denotes the tensor-matrix multiplication as follows:

$$\mathcal{L} \otimes \mathbf{T} \stackrel{\text{def.}}{=} \left( \sum_{k,\ell} \mathcal{L}_{i,j,k,\ell} \mathbf{T}_{k,\ell} \right)_{i,j}. \quad (5)$$

**Regularized Gromov-Wasserstein Discrepancy.** For computational efficiency, for large datasets, we utilize the following regularized approximation of the initial GW formulation of eq. (4)[78]

$$\text{GW}_\varepsilon(\mathbf{C},\bar{\mathbf{C}},\mathbf{u},\mathbf{v}) \stackrel{\text{def.}}{=} \min_{\mathbf{T} \in \mathcal{C}_{\mathbf{u},\mathbf{v}}} \mathcal{E}_{\mathbf{C},\bar{\mathbf{C}}}(\mathbf{T}) - \varepsilon H(\mathbf{T}), \quad (6)$$

where $\varepsilon$ is a regularization parameter and the entropy of $\mathbf{T}$ is defined as $H(\mathbf{T}) \stackrel{\text{def.}}{=} -\sum_{i=1,j=1}^{n} \mathbf{T}_{i,j}(\log(\mathbf{T}_{i,j}) - 1)$. A projected gradient descent is used to solve the non-convex optimization problem of eq. (6), where both the gradient step and the projection are computed according to the KL metric. Iterations of this algorithm are given by

$$\mathbf{T} \leftarrow \text{Proj}_{\mathcal{C}_{\mathbf{u},\mathbf{v}}}^{\text{KL}} \left( \mathbf{T} \odot e^{-\tau(\nabla \mathcal{E}_{\mathbf{C},\bar{\mathbf{C}}}(\mathbf{T}) - \varepsilon \nabla H(\mathbf{T}))} \right), \quad (7)$$

where $\tau > 0$ is a small step size, and the KL projector of any matrix K is

$$\text{Proj}_{\mathcal{C}_{\mathbf{u},\mathbf{v}}}^{\text{KL}}(\mathbf{K}) \stackrel{\text{def.}}{=} \underset{\mathbf{T}' \in \mathcal{C}_{\mathbf{u},\mathbf{v}}}{\text{argmin}} \, \text{KL}(\mathbf{T}'|\mathbf{K}). \quad (8)$$

In the special case $\tau = 1/\varepsilon$, eq. (7) becomes

$$\mathbf{T} \leftarrow \mathcal{T}(\mathcal{L}(\mathbf{C},\bar{\mathbf{C}}) \otimes \mathbf{T},\mathbf{u},\mathbf{v}). \quad (9)$$

Proof of eq. (9) is available in ref. [78]. Eq. (9) defines a computationally amenable algorithm, in which each update of $\mathbf{T}$ involves a Sinkhorn projection denoted by $\mathcal{T}$[78].

**Computational speed up.** If the loss can be written as

$$L(a,b) = f_1(a) + f_2(b) - h_1(a)h_2(b) \quad (10)$$

for functions $(f_1, f_2, h_1, h_2)$, then, for any $\mathbf{T} \in \mathcal{C}_{\mathbf{u},\mathbf{v}}$[78],

$$\mathcal{L}(\mathbf{C},\bar{\mathbf{C}}) \otimes \mathbf{T} = c_{\mathbf{C},\bar{\mathbf{C}}} - h_1(\mathbf{C})\mathbf{T}h_2(\bar{\mathbf{C}})^\top, \quad (11)$$

where $c_{\mathbf{C},\bar{\mathbf{C}}}$ is independent of $\mathbf{T}$. Proof of eq. (10) is available in ref. [78]. Eq. (11) shows that for this class of losses, we can compute $\mathcal{L}(\mathbf{C},\bar{\mathbf{C}}) \otimes \mathbf{T}$ efficiently using only matrix/matrix multiplications. In our case, the KL loss satisfies eq. (10) for $f_1(a) = a \log(a) - a, f_2(b) = b, h_1(a) = a$, and $h_2(b) = \log(b)$.

**Efficient computation of the interaction matrix.** Computation of the pairwise interaction matrix (eq. (3)) can be intensive because of the inversion of the covariance matrix. Interestingly, the formulation of the interaction strength between two genes (eq. (3)) is same as that of their partial correlation[79]. An efficient way to compute the partial correlation matrix (and thus interaction matrix) is to 1) solve the two associated linear regression problems (shown below), 2) get the residuals from the regression problems, and 3) calculate the correlation between the residuals[80]. Let us assume that $X$ and $Y$ are random variables taking real values (denoting expression levels of two genes), and let $\mathbf{Z}$ be the $(n-2)$-dimensional vector-valued random variable (denoting expression levels of all other genes). Let us also assume that $\mathbf{x}_i, \mathbf{y}_i$ and $\mathbf{z}_i, i = 1, \ldots, m$ denotes the independent and identically distributed $m$ observations from some joint probability distribution of the random variables $X, Y$ and $\mathbf{Z}$. If we want to find the relationship between the random variables through regression, we have to find the regression coefficient vectors $\mathbf{w}_X^*$ and $\mathbf{w}_Y^*$ such that

$$\mathbf{w}_X^* = \arg\min_{\mathbf{w}} \left\{ \sum_{i=1}^{m} \left( \mathbf{x}_i - \langle \mathbf{w},\mathbf{z}_i \rangle \right)^2 \right\} \quad (12)$$

$$\mathbf{w}_Y^* = \arg\min_{\mathbf{w}} \left\{ \sum_{i=1}^{m} \left( \mathbf{y}_i - \langle \mathbf{w},\mathbf{z}_i \rangle \right)^2 \right\} \quad (13)$$

with $\langle \mathbf{w},\mathbf{v} \rangle$ the scalar product between the vectors $\mathbf{w}$ and $\mathbf{v}$. The residuals can then be computed as

$$\mathbf{e}_{X,i} = \mathbf{x}_i - \langle \mathbf{w}_X^*,\mathbf{z}_i \rangle, \quad (14)$$

$$\mathbf{e}_{Y,i} = \mathbf{y}_i - \langle \mathbf{w}_Y^*,\mathbf{z}_i \rangle. \quad (15)$$

The partial correlation between $X$ and $Y$ can then be expressed as:

$$\hat{\rho}_{XY \cdot \mathbf{Z}} = \frac{m \sum_{i=1}^{m} \mathbf{e}_{X,i}\mathbf{e}_{Y,i} - \sum_{i=1}^{m} \mathbf{e}_{X,i} \sum_{i=1}^{m} \mathbf{e}_{Y,i}}{\sqrt{m \sum_{i=1}^{m} \mathbf{e}_{X,i}^2 - \left(\sum_{i=1}^{m} \mathbf{e}_{X,i}\right)^2} \sqrt{m \sum_{i=1}^{m} \mathbf{e}_{Y,i}^2 - \left(\sum_{i=1}^{m} \mathbf{e}_{Y,i}\right)^2}}. \quad (16)$$

For independent 2-way (pairwise) interactions between $X$ and $Y$ ($\mathbf{Z}$ has no effect on the interaction of $X$ and $Y$), eq. (16) can be simplified as

$$\hat{\rho}_{XY} = \frac{m \sum_{i=1}^{m} \mathbf{x}_i\mathbf{y}_i - \sum_{i=1}^{m} \mathbf{x}_i \sum_{i=1}^{m} \mathbf{y}_i}{\sqrt{m \sum_{i=1}^{m} \mathbf{x}_i^2 - \left(\sum_{i=1}^{m} \mathbf{x}_i\right)^2} \sqrt{m \sum_{i=1}^{m} \mathbf{y}_i^2 - \left(\sum_{i=1}^{m} \mathbf{y}_i\right)^2}}. \quad (17)$$

### Implementation and parameter settings

Both Python and Matlab 2020a (MathWorks Inc., Natick, MA, USA) were used to implement the genomap technique. The expression values of the genes were standardized using z-score before converting

to genomaps[81]. Sinkhorn optimization was used for datasets with more than 2500 selected genes. The value of $\epsilon$ in sinkhorn optimization was considered from 0.001 to 0.005, incrementing 0.001 at each step until convergence. For smaller datasets (gene number < = 2500), $\epsilon$ was set to zero to compute the exact transport matrix. **u** and **v** were assumed to have uniform distribution. Same values of $p$ and $q$ were selected in our analyses to make genomaps square-shaped. However, as found in our analyses, genomaps with rectangular shape ($p > q$ or $p < q$) performs similarly to the genomaps with squared shape (see supplementary Table S3). In cases, where $n < p \times q$, the excess locations in genomaps were set to zero. In this case also, the performance of genomap remains unaltered (see Supplementary Table S3).

Four state of the art cell annotation techniques: Cell-ID, Vec2-image, ACTINN and singleR and, two classification techniques: random forest[82] and SVM[44], and three boosting classification techniques: AdaBoost[83], LPBoost[84] and TotalBoost[85] were used to benchmark the proposed approach for cell recognition. All the parameters of the boosting classification techniques were set to default values of MATLAB. Cell-ID and SingleR was run using default settings instructed by the authors. Vec2image method was used with default configurations (t-SNE projection and images of $120 \times 120$ pixels). In 'Random map', the 2D images were created by randomly placing the genes in a 2D grid. PHATE and UMAP were downloaded from the Github link provided by the authors and run with the default settings. In case of supervised PHATE, LDA, Siamese network and UMAP, the labels were estimated by x-means[86] clustering technique (similarly to genoNet).

Seurat, Harmony, Online iNMF, and Cell-ID methods were installed in R using the instructions provided by the authors. The methods were run using the default configurations. For single cell data integration using genomap, the datasets were pre-aligned using canonical correlation analysis (from Seurat tool with default settings) and then genoNet is used on it to create the integrated embedding. In computing the cell-specific gene importance using Cell-ID, at first the embedding was computed for both cells and genes. As instructed by the authors, the distance between cells and genes is then computed to obtain cell-specific genes. The inverse value of the distance between a cell and the genes was used as the importance score of the genes in that particular cell. For sci-ATAC-seq dataset, the expressions of *CD3D*, *CD3E*, *CD34*, *CD79A*, and *CD79B* genes are averaged over all the cells of a specific type (such as B and T cells) to obtain its activity score. For the proposed approach, the class activation values of a gene in all the cells of a specific type are averaged to obtain the activity score. The mean activity scores from Cell-ID, our approach, and sci-ATAC-seq dataset were then normalized from 0 to 1 over the cell types. For predicting the cell labels from embeddings of Seurat, Harmony, and Online iNMF, a k-nearest neighbor algorithm with 15 neighbors were employed[34]. In Cell-ID(c), the gene signatures extracted for each cell c in a dataset D are assessed through their enrichment against the gene signatures extracted for each cell c' in a reference dataset D'. Alternatively, Cell-ID(g) takes advantage of a grouping of cells in D, where per-group gene signatures are extracted and evaluated against the gene signatures for each cell c in the query dataset D.

In the proposed genomap+genoNet analysis, the scRNA-seq data was first converted into genomaps by the maximization of system entropy without supervision. The genoNet was then trained in a supervised fashion by using the generated genomaps. To benchmark the genomap+genoNet approach, several supervised learning models (such as Cell-ID, SingleR, random forest etc) were employed. In establishing these benchmarking models, 70% and 30% of the data were selected randomly for training and testing, respectively. In all our analyses, following the common practice in bioinformatics community[69], we selected around 5–10% of the most variable genes (see Supplementary Table S6) by using 'dispersion' as the 'selection.-method' in 'findvariablefeatures' function of Seurat[87]. As demonstrated in Supplementary Fig. S21, our proposed approach outperforms all

existing methods for different number of highly variable genes (HVGs) in cell trajectory analysis. For trajectory analysis, we used the default configuration of PHATE with 100 principal components (PCs) in all analyses. However, we have also included analyses with other numbers of PCs (200 and 300) in the Supplementary Fig. S20 to demonstrate the high performance of genomap+genoNet for different numbers of PCs. For clustering analysis, we inputted the first 37 PCs into t-SNE and UMAP for generating Fig. 9, and 20 PCs into UMAP for creating Fig. 5 following the original studies. For creating Fig. 4, we used the Python codes from https://github.com/elo073/TissStab provided by the original authors. For generating the gSNE and gMAP results, we obtained the features from the second last fully connected layer of the genoNet, and then applied PCA with t-SNE and UMAP, respectively.

## Creation of distance matrix for 2D grid positions of genomaps and scaling of interaction matrix

For even number of rows/columns ($p/q$), the row (column) grid of 2D genomap starts at $-\frac{p}{2}$ ($-\frac{q}{2}$) and ends at $\frac{p}{2} - 1$ ($\frac{q}{2} - 1$). For odd $p/q$, the row (column) grid starts at $-\frac{p-1}{2}$ ($-\frac{q-1}{2}$) and ends at $\frac{p-1}{2}$ ($\frac{q-1}{2}$). The location value inside the grid is computed as $\sqrt{i^2 + j^2}$ for location ($i, j$). The matrix with the location values is then reshaped to a vector using column-wise aggregation (the vector contains the first column, then second column and so on). The pairwise Euclidean distance among the first $n$ positions of the vector is then computed to obtain the $\bar{\mathbf{C}}$ so that there is a one-to-one relationship between genes and grid locations. We fill the last $p * q - n$ positions of geomaps with zero when $n < p * q$. The interaction matrix $\rho$ is scaled from 0 to 2 by subtracting from 1 to obtain **C**. The value of **C** and $\bar{\mathbf{C}}$ for TM dataset are shown as heatmaps and tables in Supplementary Figs. S17–S19.

## Architecture and training details of genoNet

For all the classification in this paper, we used genoNet architecture with 10 layers: input layer, 1 convolutional layer with kernel size 3 and 8 channels, 2 relu layers, 3 fully connected layers, 1 dropout layer, softmax layer and classification layer (see Supplementary Table S2). We implemented genoNet in PyTorch[88] framework. Mini-batch size was set to 128 in all trainings. Shuffling was performed at every epoch in all trainings. Adam optimizer with weight decay was used for network optimization. Initial learning rate was fixed at 0.001 and weight decay rate of 0.00001 was used in all trainings. The number of neurons in last fully connected (FC) layer of the supervised genoNet was set to 100 and of the unsupervised genoNet to 512. As unsupervised genoNet is used for feature extraction, it final FC layer was set to a larger value than the supervised genoNet. Maximum epoch was set to 150 for training supervised genoNet. 30 epochs were enough for training the unsupervised genoNet. The unsupervised genoNet was trained in the same way as supervised genoNet with the data labels estimated using x-means clustering[86]. The number of initial clusters were set at 10 and the initial cluster centers were computed using k-means++[89]. For genomap+genoNet analysis, at first genomaps are created from the data and then genoNet is performed on the genomaps in supervised/unsupervised format based on the task.

## Computation of cosine similarity, accuracy, and cluster quality indices

For calculation of NMI, accuracy, cluster quality indices, we at first cluster the data into $N_g$ classes ($N_g$ is number of classes in ground truth label) by Louvain clustering technique[46,47]. In Louvain clustering, a full similarity graph is developed using the Euclidean distances among the data points. A graph clustering with random initialization is then performed to maximize the modularity of the graph[46,47]. We next find the best map of cluster labels in comparison to the ground truth labels. These cluster labels are then used to compute the indices. NMI is the

normalized mutual information[90] between the estimated labels and true labels computed following the work of Becht et al.[26]. Accuracy is the number of correctly found class labels divided by the total number of class labels. The silhouette value[91] is a measure of how similar a data point is to its own cluster compared to other clusters. The silhouette ranges from −1 to +1, where a high value indicates that the data is well-clustered. Adjusted Rand (AR) index is computed using the formula reported in the work of Hubert et al.[92]. Cosine similarity is computed by $\frac{x_1 x_2^T}{\sqrt{(x_1 x_1^T)(x_2 x_2^T)}}$, where $x_1$ and $x_2$ are two gene expression vectors of the same length and superscript $T$ denotes the transpose of the vector.

## Datasets

**Tabula Muris (TM)[22]**. This dataset represents a compendium of single-cell transcriptomic data from the model organism Mus musculus that comprises more than 100,000 cells from 20 organs and tissues of mice. The dataset reveals the gene expression in poorly characterized cell populations and enables a direct and controlled comparison of gene expression levels in cell types shared between tissues, such as T lymphocytes and endothelial cells from different anatomical locations[22]. Two distinct technologies were used for acquisition of the data: microfluidic droplet-based 3′-end counting and the full-length transcript analysis based on fluorescence-activated cell sorting. After selecting the most variable 1089 genes, we constructed genomaps of the cells and extracted different levels of features from the genomaps for high performance classification (Fig. 2).

**Ischaemic sensitivity of human tissue by single cell RNA seq[25]**. This study assesses the effect of cold ischaemic time on scRNA-seq data from human tissues using 10x Genomics 3′ scRNA-seq. The dataset is from spleen, esophagus epithelium and lung parenchyma, three tissues that had previously been reported to have differential sensitivity to ischaemia[25]. Samples were collected into Hypothermasol FRS hypothermic preservation media and dissociated fresh (as soon as possible) or at 12h, 24h, 72h post onset of cold ischaemia in the donor. Single cell and bulk RNA-seq data were generated at a series of time points and whole genome sequencing was carried out for all the donors. We used the datasets to show the efficacy of the proposed approach in cell recognition (Fig. 4).

**T cell landscape[29]**. Age-associated changes in the functionality of CD4 T cells have been linked to both declined immunity and chronic inflammation. For detailed characterization of CD4 T cell phenotypes to explain these dysregulated functional properties, Elyahu et al. used scRNA-seq and multidimensional protein analyses to profile thousands of CD4 T cells from young and old mice. It has been found from their study that the landscape of CD4 T cell subsets is very different between young and old mice. Three cell subsets, namely exhausted, cytotoxic, and activated regulatory T cells (aTregs), appear rarely in young mice and gradually accumulate with age. In our analyses, we created genomaps of the available 24,007 cells and extracted different levels of features for accurate classification (Fig. 5).

**Comprehensive single cell transcriptome lineages of a proto-vertebrate**. Studies of ascidian (sea squirt) embryos have highlighted the importance of cell lineages in animal development for over 100 years. As simple proto-vertebrates, they are also used to explore the evolutionary origins of novel cell types, such as cranial placodes and neural crest in vertebrates. To build upon these efforts, the authors in ref. [40] have determined comprehensive single cell transcriptomes of Ciona intestinalis throughout embryogenesis. 90,579 cells from 10 different developmental stages were examined, spanning the entirety of morphogenesis, from the onset of gastrulation at the 110-cell stage to the hatching of swimming tadpoles. This represents an average of over 12-fold coverage for every cell at every stage of development, owing to the small cell numbers that are a hallmark property of ascidian embryogenesis. Single cell transcriptome trajectories were used to construct "virtual" cell lineage maps and provisional gene networks for a variety of cell types, including nearly 40 different neuronal subtypes comprising the larval nervous system. In our analyses, we created genomaps of the cells and used combination of unsupervised genoNet and PHATE for cellular trajectory analysis (Fig. 8).

**Comprehensive classification of retinal bipolar neurons[43]**. The dataset analyzed in this section is from a massively parallel scRNA-seq profiled from a heterogeneous class of neurons, mouse retinal bipolar cells (BCs)[43]. The motivation of the study is to characterize and classify neuronal cells using gene expression data. From a population of 27,499 BCs, Shekhar et al. identified 15 types of neuron cells, including all types observed previously and two new types. The experimental protocol is as follows: 1) retinas from Vsx2-GFP[43] mice were dissociated, followed by fluorescence activated cell (FAC) sorting for GFP+cells. 2) Single-cell libraries were prepared using Drop-seq and sequenced. 3) Raw reads were processed to obtain the digital expression matrix (genes cells). In our analyses, we created genomaps of the neuron cells and used unsupervised genoNet for dimensionality reduction, visualization and clustering (Fig. 9).

The number of cells and genes in raw datasets and used in genomap analysis is added in Supplementary Table S6.

## Statistics & Reproducibility

No statistical method was used to predetermine sample size. No data were excluded from the analyses. The experiments were not randomized. There was no blinding. The analyses performed do not involve evaluation of any subjective matters.

## Reporting summary

Further information on research design is available in the Nature Portfolio Reporting Summary linked to this article.

## Data availability

The datasets generated during and/or analyzed during the current study are available within the manuscript and supplementary. The accession numbers of the used datasets are: Baron[36] (GSE84133), Muraro[37] (GSE85241), Segerstolpe[35] (E-MTAB-5061), Xin[38] (GSE81608), Wang[39] (GSE83139), ischaemic sensitivity[25] (PRJEB31843), TM[22] (GSE109774). T cell landscape, proto-vertebrate transcriptomics, and retinal bipolar neuron datasets were downloaded from single cell portal of Broad institute with study number SCP490, SCP454, and SCP3, respectively. Note that the user needs to register to the single cell portal of Broad Institute with an email and a password. He/she can then login into the portal with the email and password and download the datasets. All other relevant data supporting the key findings of this study are available within the article and its Supplementary Information files or from the corresponding author upon reasonable request. Source data are provided with this paper.

## Code availability

Genomap implementation is available as a web-based computational tools at http://analyxus.com/compute/genomap. Genomap implementation is also available as a Code Ocean capsule (https://doi.org/10.24433/CO.0640398.v1). Its source codes can be found at https://github.com/xinglab-ai/genomap(https://zenodo.org/badge/latestdoi/589035404)[93].

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

## Acknowledgements

We wish to thank Sarah Xing for her help in proofreading the manuscript and for useful advice in improving the manuscript. We also want to thank Hongyi Ren and Rui Yan for their useful advice in Python implementation of genomap.

## Author contributions

L.X. conceived the experiment(s), M.T.I. conducted the experiment(s), M.T.I. analyzed the results. Both authors reviewed the manuscript.

## Competing interests

A patent application based on this work has been submitted (application number: 63/479,724) by the Board of Trustees of the Leland Stanford Junior University. The names of the inventors are Lei Xing and Md Tauhidul Islam. The patent application covers all the contents of the manuscript.
