## [Peer Review File · Nature Communications]

REVIEWER COMMENTS

Reviewer #1 (Remarks to the Author):

In this version, the authors refined the notations and added some new results. In general, the quality of the paper has been improved. However, we are still not convinced of the necessity and novelty of Genomap. We also propose some analyses to validate the superiority of Genomap.

1. The authors used entropy formulas to justify why Genomap is more informative than the gene vector. However, the arguments are not convincing: the authors' 1D entropy formula assumes that genes are independent, which we know is not the case. The entropy should be defined based on genes' joint multivariate distribution, which can be estimated from the gene expression vectors and does not require the 2D-grid representation. Hence, the necessity of using the 2D-grid representation + the blackbox CNN still lacks justification. Moreover, the 2D entropy formula is unclear: how is the $p_{\{i,j\}}^k$ defined? This entropy definition is the theoretical foundation and thus needs more explanation.

2. Following the above comment, we have a suggestion for justifying the necessity of Genomap. Besides using the optimal transport to construct the map, the authors may also try a few other ways (e.g., the simplest way is a random 2D map; another way is in Comment 4). The authors can show both the entropy and the downstream analysis results will be worse than using the optimal transport.

3. The authors claim that "We have added the neural network results in the revised manuscript, where we see that genomap+genonet improves the results by more than 5% in comparison with the neural network alone." We are not sure where the result is. Based on our guess, the authors are referring to the "ACTINN" method. Please make this clear in the manuscript; for example, why the authors use the ACTINN in the comparison. In addition, it is worth noting that ACTINN always leads to 2nd best results in all comparisons. The improvement of ACTINN vs. other methods is more significant than the improvement of Genomap vs. ACTINN. Considering that ACTINN is a neural network method developed by another group and not refined by the authors (note that there are many other neural network methods for classifying cells), we are not convinced that Genomap, rather than a neural network, is the main driver for improved classification accuracy.

4. We also found a related paper published in *Briefings in Bioinformatics*: Vec2image: an explainable artificial intelligence model for the feature representation and classification of high-dimensional biological data by vector-to-image conversion: <https://academic.oup.com/bib/article/23/2/bbab584/6518046>. Although this paper used tSNE instead of the optimal transport, the goal is also to convert a gene vector into an image and then use CNN for downstream analysis. Therefore, the authors need to justify the novelty of Genomap compared to this published article.

Reviewer #2 (Remarks to the Author):

Compared to previous version, the authors have improved their manuscript significantly, especially the description of methods. However, I still have some questions about evaluations.

1. I am quite confused about how to get the visualizations of the raw data in Figs. 4, 6 and 11. These results of raw data are too bad to believe. Were all the genes used to perform PCA, UMAP or tSNE

directly? Was any dimension reduction done before UMAP or tSNE? The visualization for raw data should be performed after highly variable gene selection and dimension reduction such as PCA. In addition, in single cell RNA sequencing data analysis, PCA is just used for dimension reduction, not for visualization.

2. In Fig. 8, what is the 10 most variable genes? Why were these genes selected? Why don't just select the marker genes for these cell types such as CD3, CD79 and so on? From the Fig. 8, Cell ID got totally random results on these genes, which is unreasonable.

3. In Fig. 11, the authors should select popular single cell data analysis methods to compare with their method, such as PCA for dimension reduction (NOT for visualization), Louvain or Leiden for clustering. LDA, Siamese network, supervised UMAP, these methods are seldom used for real single cell data analysis.

Manuscript # NCOMMS-22-33829-T

Title: Cartography of Genomic Interactions Enables Deep Analysis of Single-Cell Expression Data

The authors wish to thank the editor and referees for their constructive comments. The manuscript has been extensively revised to address the questions raised by the referees, as detailed below.

Reviewer #1 (Remarks to the Author):

In this version, the authors refined the notations and added some new results. In general, the quality of the paper has been improved. However, we are still not convinced of the necessity and novelty of Genomap. We also propose some analyses to validate the superiority of Genomap.

Response: In this resubmission, we have added the analyses suggested by you to validate the superiority of Genomap for high-performance analysis of the gene expression data. We have also added additional mathematical analysis and results to show the necessity and novelty of Genomap as detailed below.

1. The authors used entropy formulas to justify why Genomap is more informative than the gene vector. However, the arguments are not convincing: the authors' 1D entropy formula assumes that genes are independent, which we know is not the case. The entropy should be defined based on genes' joint multivariate distribution, which can be estimated from the gene expression vectors and does not require the 2D-grid representation. Hence, the necessity of using the 2D-grid representation + the blackbox CNN still lacks justification. Moreover, the 2D entropy formula is unclear: how is the $p_{\{i,j\}}^k$ defined? This entropy definition is the theoretical foundation and thus needs more explanation.

Response: In this resubmission, the entropy formulation used to explain the information increase when going from 1D to 2D-grid is updated upon your suggestion. The assumption of independence of genes is now not required. By following Refs.¹⁻⁴, we provided a detailed analysis of the entropy of genomaps. Specifically, the entropy is now defined based on the joint multivariate distribution of the genes as suggested by you. It is seen from the revised formulation that the additional information (mutual entropy) in genomaps comes from the neighborhood information of the genes located in the 2D grid. Please see supplementary section 2 for details.

The spatially independent component of the entropy can be estimated from the gene expression vectors and does not require the 2D-grid representation. However, the spatial entropy (i.e. the mutual entropy) depends on the 2D grid representation. Please see supplementary section 2 and Refs. ^{1,2} for details on the procedure to compute the entropy in images.

$p_{\{i,j\}^k}$ is the joint probability of a gene located at the center of the k -th window and a gene positioned at (i,j) inside the window. Here, $k=1$ to n , n is the number of genes. We have revised the notation of the theoretical analysis of entropy in supplementary section 2 and added a detailed explanation of each of the terms.

2. Following the above comment, we have a suggestion for justifying the necessity of Genomap. Besides using the optimal transport to construct the map, the authors may also try a few other ways (e.g., the simplest way is a random 2D map; another way is in Comment 4). The authors can show both the entropy and the downstream analysis results will be worse than using the optimal transport.

Response: We have added the entropy formulation and downstream performance analysis in terms of classification accuracy for both cases (random mapping and vec2image) as suggested by you in the revised manuscript (see supplementary section 2, Figs. 4-6, S4). It is seen that the genomap achieves much better performance in classification accuracy (>5% improvement) as compared to the random mapping and vec2image. Moreover, the entropy of both cases is worse than the genomaps constructed by using the optimal transport approach as shown in supplementary section 2 (see the last paragraph).

3. The authors claim that “We have added the neural network results in the revised manuscript, where we see that genomap+genonet improves the results by more than 5% in comparison with the neural network alone.” We are not sure where the result is. Based on our guess, the authors are referring to the “ACTINN” method. Please make this clear in the manuscript; for example, why the authors use the ACTINN in the comparison. In addition, it is worth noting that ACTINN always leads to 2nd best results in all comparisons. The improvement of ACTINN vs. other methods is more significant than the improvement of Genomap vs. ACTINN. Considering that ACTINN is a neural network method developed by another group and not refined by the authors (note that there are many other neural network methods for classifying cells), we are not convinced that Genomap, rather than a neural network, is the main driver for improved classification accuracy.

Response: The reviewer is correct that we referred to ACTINN as the neural network-based method, which has been clarified in the revised manuscript. We have used this technique because it is state-of-the-art (published in 2020) and used widely (cited by more than 99 times). Moreover, the authors of the work have made the code of ACTINN public, which we have used in our benchmarking.

To prove that genomap is the main driver for improved classification accuracy, we have added results (supplementary Fig. S5) for genoNet with 1D gene expression data input (1D expression+genoNet). In this case, when compared to genomap+genoNet, the only difference is the use or not use of the genomap. It is seen that the genomap+genoNet achieves much better performance (>5% improvement in classification accuracy) than 1D

expression+genoNet. This indicates that the introduction of genomap improves classification accuracy. Furthermore, we also added results of genomap+ACTINN (with 2D input), which shows that the use of the genomap improves the classification accuracy by at least 5% in comparison with ACTINN with 1D input.

4. We also found a related paper published in *Briefings in Bioinformatics*: *Vec2image: an explainable artificial intelligence model for the feature representation and classification of high-dimensional biological data by vector-to-image conversion*: <https://academic.oup.com/bib/article/23/2/bbab584/6518046>. Although this paper used tSNE instead of the optimal transport, the goal is also to convert a gene vector into an image and then use CNN for downstream analysis. Therefore, the authors need to justify the novelty of Genomap compared to this published article.

Response:

We have added the following discussion in the revised manuscript.

“We note that there were a few attempts in the literature to convert gene expression data into images for deep learning⁵⁻⁸. In these methods, however, an image is usually made heuristically by projecting the HD data onto a 2D plane without any explicit constraint(s) on the spatial locations of the genes. As an example, In *vec2image*⁸, t-SNE (or other) embedding method is used to create the images. As a result, similar genes get clustered under the assumption of t-distribution without explicit constraints on their spatial positions for maximizing entropy. Therefore, many genes located in the outer region of the clusters have fewer neighbors than those located in the center of the cluster. The same is true for those non-clustered (i.e. isolated) genes. The reduction of the number of neighbors to many genes deteriorates the information extraction efficiency of the CNNs (because of the limited receptive field of CNN). In contrast, the genomap here is established on a solid theoretical foundation with the goal of finding the optimal spatial configurational representation of the gene-gene interactions of the system. Thus, as shown in a variety of applications in the Results section, the deep analysis of genomaps leads to discoveries of highly distinctive patterns of the complex genomic data and provides a potentially useful analytic technique.”

We have also added the results of the *vec2image* method to the revised manuscript (Figs. 4-6, S4) and showed that the genomap+genoNet outperforms the *vec2image* method significantly (the accuracy improvement is at least 5%).

Reviewer #2 (Remarks to the Author):

Compared to previous version, the authors have improved their manuscript significantly, especially the description of methods. However, I still have some questions about evaluations.

1. I am quite confused about how to get the visualizations of the raw data in Figs. 4, 6 and 11. These results of raw data are too bad to believe. Were all the genes used to perform PCA, UMAP or tSNE directly? Was any dimension reduction done before UMAP or tSNE? The visualization for raw data should be performed after highly variable gene selection and dimension reduction such as PCA. In addition, in single cell RNA sequencing data analysis, PCA is just used for dimension reduction, not for visualization.

Response: The protocol suggested by the reviewer was followed in our analysis and visualization. That is, we performed the visualization of the raw data after selection of the highly variable genes (HVGs) and dimensional reduction. We have clarified this issue in the revised manuscript. Please see the last paragraph of “Implementation and parameter settings” section.

Following the best practice in bioinformatics community⁹, only the HVGs were used in our analysis. PCA was done before t-SNE and UMAP. Based on your suggestion, we have removed the visualization results of PCA from Figures 4, 6 and 11.

2. In Fig. 8, what is the 10 most variable genes? Why were these genes selected? Why dont just select the marker genes for these cell types such as CD3, CD79 and so on? From the Fig. 8, Cell ID got totally random results on these genes, which is unreasonable.

Response: We have selected the 10 most variable genes (highly variable genes-HVGs) because these genes contain the most information about the cell types. Gene names are shown in Fig. 8. In the revised supplementary (Fig. S21), we have added results for 6 CD (cluster of differentiation) marker genes which appeared in the first 1000 HVGs, where again we see that genomap outperforms Cell-ID in computing the gene activity. Please note that expressions of CD3 and CD79 genes were not included in the original dataset provided by the authors of Ref.¹⁰. Please see Ref.¹⁰ for the description of the dataset and the gene list. We have double checked the results of cell ID for both HVGs and marker genes and found that the results on these genes are inferior to that of the genomap+genoNet. However, we believe that the index “Pearson correlation coefficient” we were using for comparing the gene activity may not be appropriate because of the assumption of Gaussian distribution of the data. In Ref.¹¹, Kowalski's analysis concludes that the distribution of Pearson correlation coefficient is not robust in the presence of non-normality. Therefore, in the revised manuscript, we used cosine similarity^{12,13} which is a popular and established index to quantify similarities between two gene expression vectors.

3. In Fig. 11, the authors should select popular single cell data analysis methods to compare with their method, such as PCA for dimension reduction (NOT for visualization), Louvain or

Leiden for clustering. LDA, Siamese network, supervised UMAP, these methods are seldom used for real single cell data analysis.

Response: Based on your suggestions, we have included the results for PCA as dimensionality reduction, Louvain for clustering in Figure 11. Once again, genomap+genoNet performs significantly better than the available methods. We have removed the visualization results of PCA from Figures 4,6 and 11 per your suggestion.

Reference:

1. Razlighi, Q. R., Kehtarnavaz, N. & Nosratinia, A. Computation of Image Spatial Entropy Using Quadrilateral Markov Random Field. *IEEE Trans. Image Process.* **18**, 2629–2639 (2009).
2. Haralick, R. M., Shanmugam, K. & Dinstein, I. Textural Features for Image Classification. *IEEE Trans. Syst. Man Cybern.* **SMC-3**, 610–621 (1973).
3. Lezon, T. R., Banavar, J. R., Cieplak, M., Maritan, A. & Fedoroff, N. V. Using the principle of entropy maximization to infer genetic interaction networks from gene expression patterns. *Proc. Natl. Acad. Sci.* **103**, 19033–19038 (2006).
4. Stein, R. R., Marks, D. S. & Sander, C. Inferring Pairwise Interactions from Biological Data Using Maximum-Entropy Probability Models. *PLOS Comput. Biol.* **11**, e1004182 (2015).
5. Bazgir, O. *et al.* Representation of features as images with neighborhood dependencies for compatibility with convolutional neural networks. *Nat. Commun.* **11**, 4391 (2020).
6. Sharma, A., Vans, E., Shigemizu, D., Boroevich, K. A. & Tsunoda, T. DeepInsight: A methodology to transform a non-image data to an image for convolution neural network architecture. *Sci. Rep.* **9**, 11399 (2019).
7. Zhu, Y. *et al.* Converting tabular data into images for deep learning with convolutional neural networks. *Sci. Rep.* **11**, 11325 (2021).
8. Tang, H., Yu, X., Liu, R. & Zeng, T. Vec2image: an explainable artificial intelligence model for the feature representation and classification of high-dimensional biological data by vector-to-image conversion. *Brief. Bioinform.* **23**, bbab584 (2022).

9. Current best practices in single-cell RNA-seq analysis: a tutorial. *Mol. Syst. Biol.* **15**, e8746 (2019).
10. Cusanovich, D. A. *et al.* A Single-Cell Atlas of In Vivo Mammalian Chromatin Accessibility. *Cell* **174**, 1309-1324.e18 (2018).
11. Kowalski, C. J. On the Effects of Non-Normality on the Distribution of the Sample Product-Moment Correlation Coefficient. *J. R. Stat. Soc. Ser. C Appl. Stat.* **21**, 1–12 (1972).
12. Cai, S., Georgakilas, G. K., Johnson, J. L. & Vahedi, G. A Cosine Similarity-Based Method to Infer Variability of Chromatin Accessibility at the Single-Cell Level. *Front. Genet.* **9**, (2018).
13. Deshpande, R., VanderSluis, B. & Myers, C. L. Comparison of Profile Similarity Measures for Genetic Interaction Networks. *PLOS ONE* **8**, e68664 (2013).

REVIEWER COMMENTS

Reviewer #1 (Remarks to the Author):

The authors have addressed our questions about if genomap is helpful. In summary, using Genomap increases $\sim 5\%$ classification accuracy (whether this improvement should be considered significant is subjective though) compared to other maps (random and vec2image). They also modified their entropy definition.

A minor suggestion: when the authors plot the comparison of different methods, the current order (alphabetical) is hard to read. We would suggest the authors reorder their results based on the performance and the relationships between methods (e.g., CNN-based method should be plotted together).

Reviewer #2 (Remarks to the Author):

The authors followed the suggested protocol and re-analyze the data and added vec2image for comparison. However, the evaluation is still not convincing to me and I still have some questions.

1. The results of raw data are much worse than the original papers. These data were processed and analyzed in their original papers just using the popular pipeline such as HVG selection, PCA and TSNE or UMAP. Why their visualizations in original papers are much better than those in the manuscript? The authors should compare their visualization results with original paper and explain why they are so different. For example, In Shekhar et al original paper published in Cell, the retinal bipolar neurons are well clustered using PCA and TSNE. But in Fig.11a, all the bipolar cells are mixed, which is unreasonable.

2. In Fig.8. the authors still selected hvgs to characterize specific cell types. The well known marker genes for these cell types are more suitable than HVGS. There will be totally different gene sets when different HVGs selection method is used. CD3 and CD79 are well known markers for T Cells and B cells. Please check the dataset if CD3E, CD3D, CD79A and CD79B are available in the gene sets. And the other CDs the author selected are not well known marker genes for any of cell types in Fig.8.

3. The cell type labels used in the manuscript are from the original paper. However, these labels are based on the cluster results and not the ground truth. So, it is hard to say that genomap+genoNet performs significantly better than other methods such as ACTINN and Vec2image as their accuracies are comparable. These small accuracy differences between these methods might come from the inaccurate labels.

Manuscript # NCOMMS-22-33829A

Title: Cartography of Genomic Interactions Enables Deep Analysis of Single-Cell Expression Data

The authors wish to thank the editor and referees for their constructive comments. The manuscript has been revised to address the questions raised by the referees and the editor, as detailed below.

Reviewer #1 (Remarks to the Author):

The authors have addressed our questions about if genomap is helpful. In summary, using Genomap increases ~ 5% classification accuracy (whether this improvement should be considered significant is subjective though) compared to other maps (random and vec2image). They also modified their entropy definition.

Response: We want to thank you for your time and constructive suggestions. We wish to note that our approach outperforms the existing neural network approaches (e.g., ACTINN and Vec2image) by 5%-10%, and the analytical approaches by 6%-60% for the analyzed datasets. Beyond the cell classification application, our approach also outperforms the existing methods in gene activity recognition (Fig. 8), single-cell data integration (Fig. 9), trajectory mapping (Fig. 10), and clustering (Fig. 11) by significant margins.

A minor suggestion: when the authors plot the comparison of different methods, the current order (alphabetical) is hard to read. We would suggest the authors reorder their results based on the performance and the relationships between methods (e.g., CNN-based method should be plotted together).

Response: We have revised our figures based on your suggestion.

Reviewer #2 (Remarks to the Author):

The authors followed the suggested protocol and re-analyze the data and added vec2image for comparison. However, the evaluation is still not convincing to me and I still have some questions.

1. The results of raw data are much worse than the original papers. These data were processed and analyzed in their original papers just using the popular pipeline such as HVG selection, PCA and TSNE or UMAP. Why their visualizations in original papers are much better than those in the manuscript? The authors should compare their visualization results with original paper and explain why they are so different. For example, In Shekhar et al original paper published in Cell, the retinal bipolar neurons are well clustered using PCA and TSNE. But in Fig.11a, all the bipolar cells are mixed, which is unreasonable.

Response: We want to thank you for your time and constructive suggestions. The reason for the difference between the results of the original paper and ours is the number of principal components (PCs) used. We used 100 PCs in the last submission to generate the results/plots, whereas the plots in the original paper used 37 PCs. In this revision, we have changed the number of PCs to that used in the original paper (i.e. 37) and displayed the corresponding results. This resolves the problem as the t-SNE plots are now similar to that in the original publication (Ref. 1-Shekhar et al.). It is seen that, regardless of the number of PCs, our proposed approach outperforms the t-SNE and UMAP in data visualization. Below, we add the figure from the original study (Ref. 1) and our Fig. 11 for comparison. We also show genomap results and the major improvements in cluster separation indicated by using arrows. Please note that the coloring scheme and initialization of our figure are different from the original one. Like the original figure in Shekhar et al., the retinal bipolar neurons are now clustered in our Fig. 11(a).

In this revision, we also revised Figs. 4 & 6, which we generated following the protocols in Refs. 2 (page 6 and <https://github.com/elo073/TissStab>) and 3. Please note that we replaced the t-SNE plots of Figs. 4 and 6 by UMAP plots in the manuscript based on the suggestions of the editor.

Below we add the UMAP visualization results from Ref. 2 and our Fig. 4 for comparison. Please note that we used different coloring schemes to better display the difference among cell classes.

Original Figure (Ref.2-page 6):

Our Figure 4: Visualization of ischaemic sensitivity dataset (left-lung, middle-esophagus, right-spleen). (a) UMAP visualizations of raw data. (b) UMAP visualizations of the genomap features at the fully connected layer of the genoNet. Major improvements in cluster separation are indicated by arrows.

2. In Fig. 8. the authors still selected hvgs to characterize specific cell types. The well known marker genes for these cell types are more suitable than HVGS. There will be totally different gene sets when different HVGs selection method is used. CD3 and CD79 are well known markers for T Cells and B cells. Please check the dataset if CD3E, CD3D, CD79A and CD79B are available in the gene sets. And the other CDs the author selected are not well-known marker genes for any of cell types in Fig. 8.

Response: We want to thank the reviewer for this suggestion. The CD3E, CD3D, CD79A, CD79B gene expressions are available in the dataset, and we have added the results for these genes in the revised manuscript (Fig. 8, shown below) replacing the previous HVGs. It is seen that both Cell-ID and our approach show higher activities of CD3D and CD3E genes in T-cells and CD79A

and CD79B genes in B-cells. It is reasonable as CD3D, CD3E and CD79A, CD79B are established markers for T-cells and B-cells, respectively ². However, our approach shows much lower activities of these genes in other cells such as Monocyte, NK cell and Pneumocyte, which aligns well with the ground truth sci-ATAC-seq data. Again, our approach significantly outperforms Cell-ID when their performances are compared with the ground truth in terms of cosine similarity.

3. The cell type labels used in the manuscript are from the original paper. However, these labels are based on the cluster results and not the ground truth. So, it is hard to say that genomap+genoNet performs significantly better than other methods such as ACTINN and Vec2image as their accuracies are comparable. These small accuracy differences between these methods might come from the inaccurate labels.

Response: We used the labels from the original paper to benchmark our method against 11 existing methods. Here, we note that the labels created in the original publication are based on both clusters and marker genes (see pages 1310 and 1311 of Ref. 1), which is a widely used technique in cell labeling. Our method outperforms ACTINN and Vec2image by 5% to 10% in cell classification. This performance improvement should be considered significant even in the presence of slight inaccuracies of the cell labeling. Here, we note that in some of our accuracy plots, the actual accuracy improvement may not look significant because of the large range of y-axis (generally 0-100). As an example, below we add Fig. 6(b), where the accuracies of our approach, Random map and Vec2Image may look comparable. However, in reality, the accuracy of our approach is 87%, whereas Random map and Vec2Image have accuracies of 81% and 82%. For convenience, we have added the source data of our accuracy plots as excel files in this resubmission. Moreover, we have reduced the range of y-axis for better visualization of the improvement in accuracy, where possible (such as Figs. 4c and 5).

The better performance of our approach is also demonstrated in better gene activity recognition (Fig. 8), single-cell data integration (Fig. 9), trajectory mapping (Fig. 10), and clustering (Fig. 11).

Reference:

1. Shekhar, K. *et al.* Comprehensive Classification of Retinal Bipolar Neurons by Single-Cell Transcriptomics. *Cell* **166**, 1308-1323.e30 (2016).
2. Madisson, E. *et al.* scRNA-seq assessment of the human lung, spleen, and esophagus tissue stability after cold preservation. *Genome Biol.* **21**, 1 (2019).
3. Elyahu, Y. *et al.* Aging promotes reorganization of the CD4 T cell landscape toward extreme regulatory and effector phenotypes. *Sci. Adv.* **5**, eaaw8330 (2019).

REVIEWERS' COMMENTS

Reviewer #2 (Remarks to the Author):

Thank the authors for addressing all my questions. A minor suggestion: In Fig8, besides the markers genes of T and B cells, the markers for other cell types should also be included.

MS#: NCOMMS-22-33829B

Title: **Cartography of Genomic Interactions Enables Deep Analysis of Single-Cell Expression Data**

The authors wish to thank the referees for their constructive comments. Below we add the reviewer's comment and our response.

Reviewer #2 (Remarks to the Author): Thank the authors for addressing all my questions. A minor suggestion: In Fig8, besides the markers genes of T and B cells, the markers for other cell types should also be included.

Response: We want to thank the reviewer for his/her time and effort in reviewing our manuscript. Based on his/her suggestion, we have added the analysis of the marker gene of hematopoietic cell, *CD34*, in the revised manuscript. Please see Fig.S8.